# Nested birth-death processes are competitive with neural networks as time-dependent models of protein evolution

**Annabel Large** [1]   **Ian H. Holmes** [1]

## Abstract

Most statistical phylogenetics analyses use simple continuous-time finite-state Markov models of point substitution to describe molecular evolution. These models enforce unrealistic assumptions like keeping sequence length fixed, ignoring insertions and deletions (indels) entirely, and making little (if any) allowance for variation in selection pressure due to interactions between amino acids. We extend the TKF92 model—the canonical hierarchical model combining an outer birth-death process for indels with an inner finite-state Markov chain for substitutions—by introducing additional nesting and latent states, allowing for structural heterogeneity. We compare these TKF92 extensions to two classes of neural seq2seq models that use evolutionary time as an input feature: the first "basic" class lacks any evolutionary modeling constraints, while the second "hybrid" class combines neural sequence embeddings with a TKF92-like likelihood function. We evaluate the per-character perplexities of all models on splits of the Pfam database of aligned protein domains. The hybrid neural models outperform their basic counterparts across all sequence embedding architectures. Furthermore, a nested TKF-based model with only 30,000 parameters is highly competitive with all neural networks (which contain tens of millions of parameters), outperforming all but two of the neural architectures tested. Taken together, our results indicate that approaches grounded in molecular evolutionary theory may provide a better fit to real alignments than unconstrained alternatives, supporting the incorporation of CTMC-based model structure within future neural phylogenetic approaches.

[1]Department of Bioengineering, University of California, Berkeley. Correspondence to: Annabel Large <annabel_large@berkeley.edu>.

*Proceedings of the 43rd International Conference on Machine Learning*, Seoul, South Korea. PMLR 306, 2026. Copyright 2026 by the author(s).

## 1. Introduction

Substitutions and insertion-deletion (indel) events can alter protein function and drive complex evolutionary phenomena. Improved understanding of sequence evolution through time-dependent statistical models benefits areas such as epidemiology, biotechnology, and population genetics.

TKF91 and TKF92 are the canonical indel models for substitutions and indel events, assuming an underlying instantaneous rate process (Thorne et al., 1991; 1992). These models take the form of hierarchically nested continuous-time Markov chains (CTMCs). At the outer level, a linear birth-death process over "links" describes changes in sequence length. At the inner level, each residue independently evolves under a finite-state CTMC. Exact finite-time solutions to these processes yield transition and emission parameters in a hidden Markov model (HMM) that defines the autoregressive likelihood $P(Z, Y \mid X, t)$. Here $Z$ is an alignment between ancestral sequence $X$ and descendant sequence $Y$, assuming a specified evolutionary drift time $t$. Other related HMM-based indel models derive principled approximations to the underlying indel process (Löytynoja & Goldman, 2005; Redelings & Suchard, 2007; De Maio, 2021; Holmes, 2020).

In reality, each residue in a protein may experience different selection pressures due to biophysical constraints. Such rate variation can be accommodated in solvable CTMC-based models by composition, albeit in limited ways. For example, heterogeneity in substitution rates and mutation biases is routinely accounted for by introducing latent position-varying parameters into the model (Yang, 1994; Le & Gascuel, 2008; Prillo et al., 2024). Indel models are similarly amenable to such extensions and compositions (Holmes, 2004).

Use of these solvable CTMC-based models has both benefits and drawbacks. Benefits include directly interpretable rate parameters that quantify evolutionary dynamics and the ability to marginalize missing or artifactual data out of the likelihood. Drawbacks include that solvable CTMCs are limited in the complexity of the phenomena they can model. For example, indel rates are restricted by local sequence context and longer-range epistatic interactions, but incorporating these into a CTMC-based model is challenging.

An alternative to deriving HMM-based solutions is to use an autoregressive neural network to approximate the same empirical finite-time distribution $P(Z, Y \mid X, t)$. Neural networks have gained traction in related areas of bioinformatics (Dotan et al., 2025; Becker & Stanke, 2024; Teng et al., 2024). Language models can capture evolutionary information from correlations in observed data (Hie et al., 2022; Brixi et al., 2025; Hayes et al., 2025) or by directly leveraging phylogenetic information (Ye et al., 2025). EvoL-STM models pairwise alignments of DNA, but without evolutionary time (Lim & Blanchette, 2020).

More recently, EvoFlows (Aziznejad et al., 2026) learns substitution and indel rates using a method based on Edit Flows (Havasi et al., 2025). Similar to molecular evolution models, EvoFlows assumes an underlying CTMC that describes how one sequence evolves into another. However, at training time, EvoFlows learns rates conditioned on both sequences of a homologous pair by maximizing a pseudo-likelihood. This learned interpolation between sequences uses a pseudotime, where $t = 1$ corresponds to arriving at the final sequence. Indel models from molecular evolution theory mechanistically describe the indel process. The true likelihood is conditioned only on the ancestral sequence, and evolutionary time is typically calibrated to the expected number of substitutions (that is, $t = 1$ means the average time for one substitution event).

Combining neural sequence embeddings with molecular evolution models would increase the latter's expressive power, as neural architectures can capture complex multi-residue interactions—both local and distal—in ways that compositions of independent, solvable local CTMCs cannot (Wilburn & Eddy, 2020).

In this work, we explore several competing approaches to modeling molecular drift of sequences by substitution and indel events. All models provide an autoregressive likelihood of the form $P(Z, Y \mid X, t)$. We curate a dataset of protein domain alignments from the Pfam database. We compare several related HMM-based models (Thorne et al., 1991; 1992; Löytynoja & Goldman, 2005; Redelings & Suchard, 2007; Holmes, 2020) based on their cross-entropy on withheld alignments. TKF92 emerges as the best of the simple HMM-based models, so we extend TKF92 via mixtures and compositions at various levels of the hierarchically nested process. The resulting HMM-based elaborations of TKF92 gain expressivity while remaining exactly solvable. Our most expressive HMM is, to our knowledge, the first HMM-based indel model to allow indel rates to depend on local sequence context (via coordination of mixture model parameters).

We also develop two classes of neural transducers (Graves, 2012; Yeh et al., 2019). One class learns site- and sample-specific substitution and indel rates for a TKF92 model,

yielding some interpretability. In contrast, the other uses generic neural network modules without any evolution-specific architectural components. Our innovation is to allow the alignment to explicitly guide cross-attention during neural network training. The motivation is that this may provide an inductive bias towards a Markovian evolutionary process suitable for phylogenetics.

Finally, we compare the HMM-based elaborations against the neural methods based on their cross-entropy on the same withheld alignments as before. Neural models that explicitly fit TKF92 indel rates always outperform their generic counterparts. Furthermore, the models developed as solvable elaborations of TKF92 prove to be competitive with neural methods while using orders of magnitude fewer parameters.

All our proposed methods are constructed to satisfy what we call the "alignment-Markovian" property. The autoregressive likelihood $P(Z, Y \mid X, t)$ is constrained to be strictly Markovian with respect to the alignment, though not necessarily with respect to the sequences themselves. HMM-based models have even stricter constraints and, thus, are alignment-Markovian by design. The main benefit of this property is that it allows the alignment to be marginalized out of the likelihood, as with TKF92 and prior HMM-based models.

## 2. Background and Prior Work

### 2.1. Alignment-Markovian Models

Let $\Omega$ be an alphabet of symbols (e.g. amino acids), and $\Omega_g = \Omega \cup \{\epsilon\}$ be this alphabet augmented with a gap symbol. A pairwise alignment (with all-gap columns excluded) may be considered a string over the alphabet of alignment columns, $\mathcal{A} = (\Omega_g \times \Omega_g) \setminus \{(\epsilon, \epsilon)\}$. Let $A \in \mathcal{A}^*$ be an alignment and let $\mathcal{X} : \mathcal{A}^* \to \Omega^*$ and $\mathcal{Y} : \mathcal{A}^* \to \Omega^*$ be projection functions extracting, respectively, the unaligned ancestor and descendant sequences $X = \mathcal{X}(A)$ and $Y = \mathcal{Y}(A)$.

The models we consider follow an *alignment-Markovian assumption*: the distribution of the next alignment column $A_k$ is restricted to depend on the full ancestral sequence, the already-emitted portion of the descendant sequence, the lapsed evolutionary time, and the gap profile of the previous alignment column $A_{k-1}$ (i.e. whether the ancestor token is a gap, the descendant token is a gap, or neither token is a gap). This yields the factorization

$$
P(A \mid \mathcal{X}(A), t) = \\
\prod_{k=1}^{|A|+1} \mathcal{T}^{Y|X}\left(A_k \mid A_{k-1}, i, j, \mathcal{X}(A), \mathcal{Y}(A_{\ldots k-1}), t\right)
$$

with $i = |\mathcal{X}(A_{\ldots k-1})|$ and $j = |\mathcal{Y}(A_{\ldots k-1})|$. The term in

the product is the *next-column conditional probability*

$$\mathcal{T}^{Y|X}(b \mid a, i, j, X, Y_{\ldots j}, t) = $$
$$\mathrm{P}(A_k = b \mid A_{k-1} = a, i, j, \mathcal{X}(A), \mathcal{Y}(A_{\ldots k-1}), t)$$

for the next alignment column $b = A_k$ given the previous alignment column $a = A_{k-1}$, current ancestor and descendant sequence positions $(i, j)$, full ancestral sequence context $X = \mathcal{X}(A)$, autoregressive descendant sequence context $Y_{\ldots j} = \mathcal{Y}(A_{\ldots k-1})$, and lapsed evolutionary time $t$.

An alignment represents an evolutionary hypothesis, rarely observable directly. For given sequences $X$ and $Y$, the most likely alignment can be inferred with the Viterbi algorithm, or the alignment can be marginalized out of the likelihood altogether using the Forward algorithm (Durbin et al., 1998).

## 2.2. Pair Hidden Markov Models

The pair HMM is a special case of an alignment-Markovian model where the next-column probability depends only on the previous alignment column and the next ancestral symbol to be aligned. We use the Moore machine framing of an HMM, where emissions are associated with states, so the next-column conditional probability factors into transition and emission probabilities

$$\mathcal{T}_{\mathrm{HMM}}^{Y|X}(b \mid a, X_{i+1}, t) = \mathcal{M}(b, X_{i+1}) T_{\tau(a), \tau(b)} E(b)$$

Here $T$ denotes the time-dependent transition matrix, defined by the indel model; $E(\ldots)$ denotes the time-dependent emissions scoring function, defined by the substitution model and equilibrium distribution; $\mathcal{M}(b, X_{i+1})$ is an indicator function that returns 1 if the given ancestor token is contained in the alignment column, and 0 otherwise; and $\tau(\ldots)$ is the state type of the alignment column, i.e. its gap profile. The allowable state types are match (M) for columns containing two non-gap symbols $(x, y)$; insert (I) for ancestral gaps aligned to descendant symbols $(\epsilon, y)$; delete (D) for ancestral symbols aligned to descendant gaps $(x, \epsilon)$; and sentinel tokens at sequence end, Start (S) and End (E).

## 2.3. F81 Point Substitution Models

TKF92-based models rely on point substitution models to describe evolution at individual sites of the sequence. We use the F81 substitution model, a simple model parameterized by its equilibrium distribution (Felsenstein, 1981).

Let $s(t) \sim \mathrm{Subst}(Q, \pi)$ denote the point substitution process, where $s(t)$ is one residue at time $t$, $\pi$ is the equilibrium distribution, and $Q$ is the rate matrix (calculated from equilibrium distribution and exchangeabilities). The distribution of descendant symbols in match columns is $\mathrm{P}(y \mid x, t) = \exp(Qt)_{xy}$, where $\exp(\ldots)$ denotes the matrix exponential. The distribution of descendant symbols in insert columns is given by the equilibrium distribution.

With respect to the next-column conditional probability, the emissions scoring function $E(b)$ returns the emission probability corresponding to the state type of column $b$. In practice, we also marginalize over a mixture of four freely-fit (i.e. not Gamma distributed) substitution rate classes (omitted from derivations in this work; refer to (Yang, 1994) for background).

## 2.4. TKF91 and TKF92 Indel Models: Birth-death Processes over Links and Fragments

A point substitution process alone cannot generate sequences with different lengths. TKF91 was the first indel model to describe both point mutations and single-residue indel events in a time-dependent manner (Thorne et al., 1991). However, under TKF91, estimates of indel event counts (and, hence, indel rates) tend to be inflated, since the model only permits one residue to be inserted or deleted per event. Consequently, alignments inferred under TKF91 tend to be unrealistic, with single-character indels scattered all over the sequence instead of being concentrated in a few multi-character indel events. The TKF92 model improves on TKF91 by allowing multiple residues to be inserted or deleted via a single event (Thorne et al., 1992). The nested structure of the TKF models is:

**Outer model:** The outer model is a linear, continuous-time birth-death process with immigration, where "mortal links" are born with insertion rate $\lambda$ and die with deletion rate $\mu$. There is also one "immortal link" that can never die. Each link is associated with a nested process, which evolves independently throughout the life of the link. Let $S(t) \sim \mathrm{Links}(\mathrm{M}; \lambda, \mu)$ denote the hierarchically structured *links process*, where $S(t)$ is a sequence at time $t$ and M is the inner stochastic process acting at each mortal link.

**TKF91 inner model:** In TKF91, each mortal link represents a single residue evolving according to the finite-state CTMC

$$S(t) \sim \mathrm{Links}(\mathrm{Subst}(Q, \pi); \lambda, \mu)$$

for some substitution rate matrix $Q$ and equilibrium $\pi$.

**TKF92 inner model:** In TKF92, each mortal link represents a multi-residue fragment with independently evolving sites. The fragment $s'(t)$ at time $t$ is drawn from a *fragment process* $s'(t) \sim \mathrm{Frag}(\mathrm{M}; r)$ as follows. First a fragment length $K \sim \mathrm{Geometric}(r)$ (with $K \geq 1$) is sampled. The stochastic process corresponding to each character in the fragment is then drawn from the substitution model M. The TKF92-distributed "sequence" of fragments is

$$S(t) \sim \mathrm{Links}(\mathrm{Frag}(\mathrm{Subst}(Q, \pi); r); \lambda, \mu)$$

Both TKF91 and TKF92 are amenable to exact analysis, due to the tractability of the underlying birth-death model (Thorne et al., 1991; 1992). With respect to the next-column

conditional probability, closed-form solutions to the TKF processes determine the transition probabilities between column state types, which populate the transition matrix $T$.

## 2.5. General Geometric Indel Process

The TKF92 model was introduced by Thorne et al. to remedy perceived problems with the TKF91 model: namely, its failure to model multi-character indel events. TKF92 solves this problem at the cost of introducing latent information into the sequence in the form of fragment boundaries. As a result, the underlying state space of TKF92 is not the set of sequences over the residue alphabet, but the set of sequences of multi-character fragments. Fragment boundaries are never observed, as they are technical conveniences that have no material interpretation. They must be imputed or (preferably) marginalized in any practical analysis. These closed-form transition probabilities of TKF91 and TKF92 inherently include this marginalization.

This issue with TKF92 led several previous researchers to investigate CTMC models whose state consisted solely of a character sequence, with no hidden fragment structure. The canonical example is the General Geometric Indel (GGI) process (De Maio, 2021). This process assumes that indels occur at uniform random positions in the sequence and have geometrically distributed lengths, which corresponds to the maximum entropy assumptions for expected indel rates and lengths. The TKF92 model may be considered as an approximation to the GGI process; for the special case where indels are restricted to single residues, the GGI process reduces exactly to the TKF91 model.

Unlike TKF92 or TKF91, the finite-time transition probabilities of the GGI process have no closed-form solution. However, it is straightforward to simulate trajectories from this process, and one can then compare the observed gap length distributions from these simulations with various approximations and heuristics. Several such approximations have been proposed and evaluated on simulated data (Knudsen & Miyamoto, 2003; Löytynoja & Goldman, 2005; Redelings & Suchard, 2007; De Maio, 2021; Holmes, 2020). Of these approaches, the renormalized ODE approach of (Holmes, 2020), which we refer to as H20, most accurately approximates the distributions of gap lengths observed in GGI-based simulations; it is closely followed by TKF92.

As part of our efforts to develop the theory of indel models beyond TKF92, we assessed these approaches not just as approximations to the idealized GGI process, but as explanatory models of real protein alignments. Along with H20, TKF91, and TKF92, we considered two other models, LG05 (Löytynoja & Goldman, 2005) and RS07 (Redelings & Suchard, 2007), both of which can be characterized as informed guesses at closed-form solutions.

## 2.6. Mixture of Site Classes

Using mixture distributions is the standard approach to modeling heterogeneous selection pressures across sites (Yang, 1994; Le et al., 2008). Instead of being drawn from one point substitution process, each site is drawn from a categorical mixture of substitution processes.

$$\text{MixSites}(u, Q_c, \pi_c) \equiv \text{Mix}_{c \sim u} \left( \text{Subst}\left( Q_c, \pi_c \right) \right)$$

where $\text{Mix}_{k \sim p}(F(\theta_k))$ denotes a mixture of component models $F(\theta_k)$ with component weights $p_k$ and categorical class label $k$. Here, $c \sim u$ denotes site class.

By model design, the ancestral and descendant residues within an alignment column must share the same categorical label; site classes remain fixed and do not evolve. In order to evaluate alignment likelihoods, the unobserved latent class labels must be marginalized out. Conditioning on the ancestor sequence can make this marginalization cumbersome, so we instead evaluate the jointly-normalized likelihood $P(A, \mathcal{X}(A) \mid t)$. The autoregressive decomposition uses the *next-column joint probability* $\mathcal{T}_{\text{HMM}}^{XY}(b \mid a, X_{i+1}, t)$. This likelihood has the same factorization as $\mathcal{T}_{\text{HMM}}^{Y|X}(b \mid a, X_{i+1}, t)$, but the transition and emission probabilities include additional terms to account for the ancestral sequence.

# 3. Proposed Models

## 3.1. TKF-based Mixtures of Processes

### 3.1.1. MIXTURE OF FRAGMENT CLASSES

Instead of drawing from one fragment process, each TKF92 fragment is drawn from a categorical distribution of fragment processes. Each component fragment process also contains its own mixture over point substitution processes.

$$\begin{aligned}\text{MixFrag}(w, r_f, \theta_{cf}) \equiv \\ \text{Mix}_{f \sim w}(\text{Frag}(\text{MixSites}(u_f, Q_{cf}, \pi_{cf}); r_f))\end{aligned}$$

where $f \sim w$ is the fragment class label and $\theta_{cf} = (u_f, Q_{cf}, \pi_{cf})$ is the collection of parameters for the inner-level substitution processes. When evaluating the joint likelihood, latent site and fragment class labels appear as state indices in an expanded pair HMM and can be marginalized out with the forward algorithm.

### 3.1.2. MIXTURE OF DOMAIN CLASSES

Here, we nest a mixture of TKF92-based models *inside* a TKF91 links model.

**Outer model:** The outer model is a TKF91 birth-death process where outer-level mortal links are born with insertion rate $\lambda_0$ and deletion rate $\mu_0$.

**Inner model:** Each outer-level mortal link is associated with a subsequence of multi-residue fragments, which is

generated by an inner-level TKF92 model. First, a TKF92 model is sampled from a categorical mixture; let $n \sim v$ be its domain class label. Next, an inner-level mortal link is either inserted with rate $\lambda_n$ or deleted with rate $\mu_n$. The inner-level fragment length is then drawn from a mixture of fragment processes, and each site in the fragment is drawn from a mixture of substitution models

$$\text{MixDom}(v, \lambda_0, \mu_0, \phi_{cfn}) \equiv$$
$$\text{Links}(\text{Mix}_{n \sim v}(\text{MixFrag}(w_n, r_{fn}, \theta_{cfn}); \lambda_n, \mu_n); \lambda_0, \mu_0)$$

where $\phi_{cfn} = (w_n, r_{fn}, \theta_{c,f,n}, \lambda_n, \mu_n)$ are the parameters for all lower-level mixture models. All latent classes manifest as additional states in the pair HMM and are marginalized out of the joint likelihood with the forward algorithm.

## 3.2. Basic Neural Model

All previously described methods define an instantaneous evolutionary model and then attempt to solve the model's finite-time transition probabilities (exactly or approximately). As an alternative, the *Basic neural model* is a kind of neural transducer that generates the aligned descendant sequence autoregressively, with the ancestral sequence and lapsed evolutionary time supplied as input features. We hypothesize that this model provides a reasonable approximation to the corresponding finite-time transition probabilities, despite the fact that this model is not based on a continuous-time Markov chain. Neural models do not contain latent class labels, so they are trained to maximize the conditional likelihood $P(A \mid \mathcal{X}(A), t)$.

### 3.2.1. CONCISE ALPHABET OF ALIGNMENT COLUMNS.

There exist $(|\Omega| + 1)^2$ possible alignment columns. However, only a subset of columns can have finite probability after conditioning on an ancestral sequence. Let $\Omega_{\text{aug}} = (\Omega \times \{\text{M}, \text{I}\}) \cup \{\epsilon, \text{E}\}$ be the concise alphabet that represents the possible columns after specifying an ancestral token. Let $\mathcal{U}(x)$ be the $|\mathcal{A} \cup \{\text{E}\}| \times |\Omega_{\text{aug}}|$ matrix that maps probabilities over the concise column alphabet to probabilities over the full column alphabet, conditioned on the ancestral token.

### 3.2.2. ALIGNMENT LIKELIHOOD.

In the Basic neural model, the next-column conditional probability $\mathcal{T}_{\text{neu}}^{Y|X}(b \mid a, i, j, X, Y_{...j}, t)$ depends on position-specific embeddings from two neural networks:

- $B : \Omega^{L_X} \to \mathbb{R}^{L_X \times H}$ creates embeddings for an ancestral sequence of length $L_X$, with embedding dimension $H$. Each embedding vector $B(X)_i$ uses context from the entire ancestral sequence $X$.

- $D : \Omega^{L_Y} \to \mathbb{R}^{L_Y \times H}$ creates embeddings for a de-

scendant sequence of length $L_Y$. This embedding has causal context with respect to the descendant sequence: $D(Y)_j \equiv D(Y \otimes m_{\leq j})_j$ where $m_{\leq j}$ masks all positions after $j$.

The next-column conditional probability is generated by

$$z_{ij} = F(B(X)_{i+1}, D(Y)_j, \text{onehot}(\tau(a)), [t])$$
$$\mathcal{T}_{\text{neu}}^{Y|X}(b \mid a, i, j, X, Y_{...j}, t) =$$
$$(\mathcal{U}(X_{i+1}) \cdot \text{softmax}(z_{ij}))_b$$

Function $F$ transforms the input features into logits $z_{ij}$. Specifically, sequence embeddings at $(i+1, j)$ are concatenated, layer normalized, and further concatenated with the previous column's one-hot encoded state type and a single-element vector containing the lapsed evolutionary time. These column-wise embeddings are passed into a two-layer feedforward network with dropout, and the final logits are transformed into probabilities over the concise alphabet using softmax.

### 3.2.3. SEQUENCE EMBEDDING MODELS

We evaluate three architectures for sequence embedding models $B$ and $D$: a residual CNN (He et al., 2016; LeCun et al., 1989), an LSTM (Hochreiter & Schmidhuber, 1997) and a Transformer with rotary positional embeddings (RoPE) (Vaswani et al., 2017; Su et al., 2024). The residual CNN consists of a pre-norm residual block with a convolution layer, layer normalization, SiLU activation, and dropout (Ba et al., 2016; Elfwing et al., 2017; Srivastava et al., 2014). The Transformer follows a standard pre-norm architecture with a self-attention block, a two-layer multi-layer perceptron (MLP), residual connections around the two blocks, layer normalization, SiLU activation, and dropout.

## 3.3. Neural TKF Model: a Hybrid Approach

We now combine instantaneous evolution processes with neural sequence embeddings. The *Neural TKF model* assumes a distinct TKF92+F81 model at every alignment column. The evolutionary model parameters are generated by neural networks from the available alignment-Markovian context. First, logits are generated by

$$(z_{ij}^\lambda, z_{ij}^\mu, z_{ij}^r) = G(B(X_{i+1}), D(Y_j), \text{onehot}(\tau(a)))$$
$$z_{ij}^\pi = W(B(X_{i+1}), D(Y_j))$$

where $G$ and $W$ each transform their input features into logits $z_{ij}^\theta$, with $\theta = (\lambda, \mu, r, \pi)$ being parameters of a TKF92+F81 model. As before, sequence embeddings at $(i+1, j)$ are concatenated and layer normalized. $G$ also concatenates the one-hot encoded state type of the previous column, but $W$ omits this context. The resulting

column-specific embeddings are passed into separate two-layer feedforward networks with dropout to generate logits $z_{ij}^{\theta}$. These logits are transformed into evolutionary model parameters by activation functions that ensure valid parameter values. These parameters yield site- and sample-specific TKF92+F81 transition and emission probabilities.

The Neural TKF next-column conditional probability is similar to that of the pair HMMs.

$$\mathcal{T}_{\text{NTKF}}^{Y|X}\left(b \mid a, i, j, X, Y_{...j}, t\right) = \mathcal{M}(b, X_{i+1}) T_{\tau(a)\tau(b)}^{ij} E^{ij}(b)$$

with a unique transition matrix $T^{ij}$ and emission scoring function $E^{ij}(b)$ for every position $(i, j)$.

# 4. Methods

## 4.1. Dataset Curation

Pairwise alignments are curated from Pfam 36.0 (Sonnhammer et al., 1997), specifically the Pfam-A seed alignments and phylogenetic trees. First, we discard: sequences repeated within an MSA (keeping only the first instance); sequences repeated across MSAs (i.e. across Pfams); peptides shorter than 10 amino acids; sequences containing nonstandard characters (B, O, U, X, Z); and Pfams with only one member. Twelve alignments were missing corresponding trees; these are imputed with FastTree 2.1.11 (Price et al., 2010), the same program used by Pfam (Finn et al., 2014).

We extract pairwise alignments from MSAs following the method described in (Prillo et al., 2023). Briefly, the two closest leaves in the tree are pruned and paired together. This process is repeated until the entire MSA is exhausted. If a single sequence remains unpaired, it is removed from the dataset. Sequence pairs are not realigned. Although observed alignments can never be assumed to be true, our models treat alignment as a latent variable that can be marginalized out of the likelihood. Since we assume time-reversible models, each pairwise alignment yields two samples: one with the first sequence as the ancestor and the second as the descendant, and another sample with the roles reversed. Intermediate nodes do not need to be considered; by Felsenstein's pulley principle (Felsenstein, 1981), we may re-root such that either sequence can be treated as the ancestor. After postprocessing, fewer than 2% of pairwise alignments exceed 512 columns, and those that do have exceptionally many columns (up to 2,500). These longer alignments are removed to enable faster model prototyping.

In summary, we retained 19,909 Pfams and extracted 600,782 pairs of sequences to obtain a final training dataset of 1,201,564 pairwise alignments.

## 4.2. Train-dev-test Partitions

Pfam clans and the remainder of families without clan labels are randomly assigned to one of ten splits. Once assigned, all the pairwise alignments of the given clan or family must be placed in the same split. This prevents prior knowledge of sequence homology from leaking to other splits. Assignment is balanced such that each split contains an approximately equal number of pairwise alignments. From these splits, three different permutations of train, dev, and test sets are created. Each partition has a 70:10:20 train-dev-test ratio.

## 4.3. Experiments

Simple indel models and neural models are trained to minimize the average of the conditional negative log-likelihood (NLL) over all alignments in the training set $\mathcal{D}$:

$$\mathcal{L} = -\frac{1}{|\mathcal{D}|} \sum_{A \in \mathcal{D}} \log P\left(A \mid \mathcal{X}(A), t\right)$$

Hierarchical mixture models are trained to minimize the average jointly-normalized NLL. By subtracting the marginal NLL, $\log P\left(\mathcal{X}(A)\right)$, from the joint NLL after training, hierarchical mixtures can be directly compared to neural models.

Training is repeated across three distinct train-dev-test partitions. Models are evaluated using the total NLL on the held-out test set, averaged over the three partitions. We also measure the exponentiated cross-entropy (ECE), which is computed by (i) normalizing each alignment's NLL by the length of its descendant sequence, (ii) averaging these length-normalized values across all samples, and (iii) exponentiating the resulting average. This metric is essentially a per-column perplexity.

Our first investigation evaluates indel models that approximate or are special cases of the GGI process: TKF91, TKF92, LG05, RS07, and H20 (Thorne et al., 1991; 1992; Löytynoja & Goldman, 2005; Redelings & Suchard, 2007; Holmes, 2020). The observed amino acid frequencies are used as the equilibrium distribution, and substitution probabilities are calculated with the F81 model. Branch lengths are provided by Pfam trees. We also briefly explored using the GTR substitution model with LG08 exchangeabilities (GTR-LG08) (Lanave et al., 1984; Le & Gascuel, 2008) and marginalizing over a grid of proposed times (where time is treated as a latent variable). Both yielded consistent conclusions (see Appendix D).

Moving forward, we use the observed amino acid frequencies as the equilibrium distribution only if there is one substitution process. Otherwise, distributions are fit during training. Our second investigation evaluates hierarchical mixtures of site, fragment, and domain classes and compares their Akaike information criterion (AIC) (Akaike,

1974) gains relative to the reference TKF92+F81 model. The number of mixture components is held constant across the lower hierarchical levels. For example, a model with two domain classes also includes two fragment classes and two site classes, yielding $2^3 = 8$ distinct combinations of labels. The numbers of mixture components tested are: 2-5, 10, 20, 30, 175, 500, and 900 for the mixture of site classes; 2-5, 10, 20, and 30 for the mixture of fragment classes; and 2-5 and 10 for the mixture of domain classes.

Our final investigation compares neural methods against the best mixture-based models. Initial trials tested six configurations of architectures formed by combining one of three sequence embedding architectures (Residual CNN, LSTM, and Transformer) with one of two prediction heads (Basic neural and Neural TKF). For each configuration, ancestor and descendant embedders share identical architectures and hyperparameters, except where necessary to enforce causal descendant sequence context.

We optimize a Transformer-based embedding architecture through hyperparameter sweeps while keeping the Neural TKF prediction head fixed. The sweeps vary the embedding dimension, number of Transformer blocks, number of self-attention heads, dropout rate, global learning rate, and weight decay. Performance is evaluated using the total NLL on a single dev set. The final Transformer-based model is called the "6-block" model, which reflects the optimal number of blocks found during the sweep. The initial Transformer configuration is called the "1-block" model. The same sequence embedding architecture is used for both the Neural TKF and Basic neural models, although only the former is explicitly optimized. We briefly explored alternative architectures for the Neural TKF prediction head (with the sequence embedders fixed), but the choice of sequence embedding architecture had more effect on model fit.

All models are trained using mini-batch stochastic gradient descent with the Adam optimizer (Kingma & Ba, 2017) on NVIDIA GeForce RTX 2080 Ti and RTX A6000 GPUs. Details and training times can be found in Appendix C.

## 5. Results

### 5.1. Evaluation of Approximations to the GGI Process

Table 1 compares basic indel models and reports the total test set NLL, averaged over three train-dev-test partitions. Because TKF91 overestimates the number of indel events, it infers inflated indel rates and yields the poorest likelihoods. LG05 and RS07 make similar approximations to the finite-time transition probabilities. They have comparable performance and intermediate placement in the evaluation table, matching results from simulation studies. On average, TKF92 demonstrates a slightly better fit to real alignments than H20, despite H20 proving better for simulated gap

*Table 1.* (**Sec. 5.1**) **Comparing indel models by NLL of the held-out test set; lower NLL and ECE are better:** "Total NLL ($\times 10^6$)" is the NLL of the test set, summed over samples. "Average NLL" is the NLL averaged over samples. Metrics for the best indel model, TKF92, are **bolded**.

| INDEL MODEL | TOTAL NLL ($\times 10^6$) | AVERAGE NLL | ECE |
|---|---|---|---|
| TKF92 | **70.30** | **281.203** | **7.1398** |
| H20 | 70.35 | 281.416 | 7.1530 |
| LG05 | 72.46 | 289.861 | 7.5423 |
| RS07 | 72.47 | 289.887 | 7.5434 |
| TKF91 | 73.87 | 295.506 | 7.7942 |

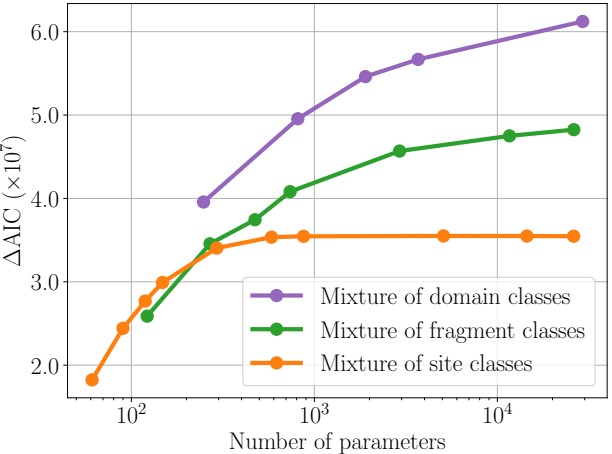

*Figure 1.* (**Sec. 5.2**) **TKF-based mixtures of processes: gain in AIC over TKF92+F81; higher is better:** Each dot corresponds to a single trial with a fixed number of mixture components, detailed in Section 4.3. The x-axis shows the log-scaled number of model parameters and the y-axis shows gain in AIC over the reference model, TKF92+F81. Each curve asymptotically approaches (or is projected to approach) a limit on the performance gains obtained from increasing the number of components.

profiles.

These initial investigations informed our subsequent model design and evaluations. Treating time as a latent variable and marginalizing it out increased computational cost with negligible effects on likelihood or indel model rankings. Therefore, we proceeded to evaluate likelihoods using Pfam branch lengths. Since TKF92 has a closed-form solution, is straightforward to elaborate further, and fits observed alignments best, it serves as the basis for our subsequent attempts to describe indel evolution more realistically. We accept the latent fragment boundaries inherent to TKF92 as a minor technical inconvenience that does not hurt performance.

*Table 2.* (**Sec. 5.3**) **Comparing neural models against TKF-based hierarchical mixture models by NLL of held-out test set; lower NLL and ECE are better:** "Number of parameters" is the number of trainable parameters in the model. $k_n$ and $k_f$ are the number of domain and fragment classes, respectively. "Conditional column frequencies" refers to alignment likelihoods calculated according to a simple $21 \times 21$ frequency matrix built from observed counts (treating gap as a member of the sequence alphabet). Other column headings match those in Table 1. Metrics for the best model, Neural TKF with 6-block Transformer sequence embedders, are **bolded**. We also mark the 10-component mixture of domain classes with a star ($\star$) for its competitive performance with larger neural networks.

| MODEL TYPE | MODEL SUBTYPE | NUMBER OF PARAMETERS | TOTAL NLL ($\times 10^6$) | AVERAGE NLL | ECE |
|---|---|---|---|---|---|
| NEURAL TKF (SEC. 3.3) | 6-BLOCK TRANSFORMER | 43.55 M | **61.72** | **246.867** | **5.5169** |
| NEURAL TKF | LSTM | 30.08 M | 61.85 | 247.388 | 5.5340 |
| NEURAL TKF | RESIDUAL CNN | 28.53 M | 62.47 | 249.864 | 5.6258 |
| NEURAL TKF | 1-BLOCK TRANSFORMER | 29.78 M | 62.99 | 251.962 | 5.6983 |
| BASIC NEURAL (SEC. 3.2) | LSTM | 27.00 M | 62.77 | 251.068 | 5.6877 |
| BASIC NEURAL | 6-BLOCK TRANSFORMER | 42.04 M | 62.82 | 251.282 | 5.6992 |
| BASIC NEURAL | RESIDUAL CNN | 26.47 M | 63.17 | 252.661 | 5.7455 |
| BASIC NEURAL | 1-BLOCK TRANSFORMER | 26.87 M | 64.04 | 256.156 | 5.8817 |
| MIXTURE OF | | | | | |
| . . . DOMAIN CLASSES (SEC. 3.1.2) | 10 COMPS. | $\star$ 29.23 K | $\star$ 62.14 | $\star$ 248.541 | $\star$ 5.7013 |
| . . . FRAGMENT CLASSES (SEC. 3.1.1) | 30 COMPS. ($k_n = 1$) | 26.16 K | 63.87 | 255.479 | 5.9602 |
| . . . SITE CLASSES (SEC. 2.6) | 30 COMPS. ($k_n = k_f = 1$) | 873 | 65.00 | 260.028 | 6.1238 |
| TKF92+F81 (SEC. 2.4) | N/A | 11 | 70.30 | 281.203 | 7.1398 |
| TKF91+F81 (SEC. 2.4) | N/A | 10 | 73.87 | 295.506 | 7.7942 |
| CONDITIONAL COLUMN FREQUENCIES | N/A | 0 | 78.41 | 306.981 | 7.2257 |

## 5.2. Evaluation of TKF-based Hierarchical Mixtures

In Figure 1, the gain in AIC (Akaike, 1974) is plotted for all hierarchical mixture models (compared to TKF92+F81). This figure is generated from one train-dev-test partition, but trends are consistent across all three. Results demonstrate that increasing the number of site class components improves model fit, as compared to assuming only one substitution process; this is consistent with prior work (Yang, 1994; Le et al., 2008). However, across all latent class types, increasing the number of components yields diminishing improvements in model fit. In particular, as the number of components grows, models exhibit component collapse and redundancy. Better fit is achieved only by designing higher levels of process, as in our MixDom and MixFrag models.

This behavior aligns with our expectations about how heterogeneous evolutionary forces are structured. Coordinating substitution mixtures within each fragment class reflects the expectation that residues collocated in sequence (and hence in space) are often subject to similar substitution pressures. Synchronizing mixtures of fragment processes (along with fragment-level indel rates) within each domain class reflects the fact that larger structural or functional regions have specific tolerances to indel events, in addition to known substitution biases. For example, the hydrophobic core of a globular protein tends to have poor tolerance for big insertions or hydrophilic amino acids.

Practically, we never observed overfitting during model training, even for MixDom. Although substantially larger

mixtures would likely overfit, increasing to such scale would be unjustifiable due to diminishing returns in performance. This suggests that the largest hierarchical mixtures have reached or are approaching their respective capacity ceilings.

## 5.3. Evaluation of Neural and Mixture-based Models

Table 2 compares all neural models against the best-performing hierarchical mixtures and reports the total test set NLL and related metrics, averaged over three train-dev-test partitions. "Conditional column frequencies" refers to likelihoods calculated according to a $21 \times 21$ (i.e. 20 amino acids plus the gap character) frequency matrix built from observed counts in the training set. This baseline likelihood also assumes geometrically-distributed alignment lengths.

By all metrics and for all sequence embedding architectures, the hybrid Neural TKF models consistently outperform the Basic neural models. That is, explicitly incorporating the inductive bias of the TKF indel model benefits every neural network evaluated here. Overall, the best-performing model is the Neural TKF with 6-block Transformer sequence embedders.

The 10-component MixDom model ranks third by total NLL and seventh by exponentiated cross-entropy (ECE), a surprising feat for a model with three orders of magnitude fewer parameters than the neural methods. The 30-component mixture of fragment classes, an even smaller pair HMM, also outperforms the Basic neural (1-block Transformer)

model. These results demonstrate that smaller hierarchical mixtures can be competitive with and even outperform neural methods, despite their small parameter set and low capacity ceiling.

In Appendix F, we further stratify results based on branch length, sequence identity between the ancestor and descendant, and gap content of the pairwise alignment. At the extreme ends of each (i.e. for the most distant pairs, most dissimilar sequences, and gappiest alignments), MixDom outperforms all neural models. While Neural TKF (6-block Transformer) is the best-performing model based on the entire test dataset, this stratification reveals that it generalizes worse to more distant pairs than MixDom.

Taken together, these results suggest that incorporating the inductive bias of evolutionary models (whether in hierarchical mixtures or hybrid architectures) improves model fit to Pfam pairwise alignments. As measured by likelihood-based metrics, the Neural TKF (6-block Transformer) model is superior. Future directions for model development include incorporating pre-trained protein language model embeddings (Hayes et al., 2025). The 10-component MixDom model should not be discounted. Despite its compact size, this model is competitive with neural methods evaluated here, and it generalizes better to more distant alignments. An additional comparison between all models and Progen 2 (Nijkamp et al., 2023), a modern large protein language model, can be found in Appendix E.

## 6. Discussion

In this work, we curated a training dataset of pairwise alignments and branch lengths from Pfam, and we re-evaluated indel models that approximate or reduce to the GGI indel process. Holmes (Holmes, 2020) demonstrated that H20 was better than TKF92 at recovering gap profile distributions, as simulated by the GGI process. However, our results show that TKF92, not H20, is a better fit for real data. This is surprising but not entirely inexplicable; simulated gap profiles are known to exhibit different length distributions than empirical ones (Wygoda et al., 2024).

We extended TKF92 into increasingly nested hierarchical mixtures of processes and designed two classes of comparable neural models. Overall, the hybrid approach of combining neural architectures with molecular evolution theory resulted in a model that best fit our alignments (i.e. the Neural TKF with Transformer-based sequence embedders). Across all architectures, hybrid neural models with the inductive bias for evolutionary models consistently outperformed their counterparts that lacked this bias. Similar conclusions have been drawn in several prior studies. Prillo et al. (Prillo et al., 2024) found that fitting a point substitution model to every alignment column produced a

variant effect prediction model that outperformed neural methods. Hsu et al. (Hsu et al., 2022) also found that an augmented Potts model (Levy et al., 2017), which models co-evolutionary signals, was competitive with the neural protein language models of the time at predicting function from sequence.

When fitting hierarchical mixtures, adding hierarchical levels gave a greater boost to model fit than increasing the number of components in any single level. These mixtures are competitive with neural methods at fitting Pfam alignments, despite having three orders of magnitude fewer parameters. To explain this, we note that selection is heterogeneous across residues and context-dependent due to biophysical constraints. Neural architectures incorporate complex inter-residue interactions by design. Our results suggest that adding hierarchical mixtures to evolutionary models allows them to capture progressively more complex and realistic selection forces. In particular, the mixture of domains model introduces latent domain classes that contain coordinated fragment-level indel rates and substitution processes. Thus, surrounding sequence context can indirectly influence indel dynamics, despite the two being independent processes. This may explain why the performance of the mixture of domain classes is comparable to much larger neural networks. However, the current largest MixDom model is likely close to its capacity ceiling.

A major advantage of HMM-based approaches lies in their amenability to exact statistical manipulation. Recent work has derived sufficient statistics for the TKF92 process, extended this hierarchical mixture modeling approach to explicitly incorporate co-evolutionary signals (based on Potts model couplings), and outlined a method for composing such models on phylogenetic trees (Large & Holmes, 2026). Achieving analogous tractability with the neural seq2seq models would be considerably more challenging; one conceivable route would be a variational approach in which posteriors from a TKF-like model serve as a structured approximate family, but this remains to be developed. More broadly, the complementary strengths of mechanistic and neural approaches suggest many productive directions: richer latent-state structure in the HMM family, tighter integration of alignment uncertainty into neural training, and hybrid architectures in which a TKF-derived prior regularizes a neural likelihood are all natural extensions of the present work.

An inherent limitation of all evolutionary models is the assumption of time reversibility, which may not be biologically realistic. However, time reversibility enables tractable computational methods and is a standard assumption across all of phylogenetics.

In conclusion, CTMC-based models remain a relevant framework for describing molecular evolution, even in the age of deep learning.

## Acknowledgments

We thank Jeffrey Thorne, Yun Song, Sebastian Prillo, and Antoine Koehl for helpful discussions and feedback. We also thank the anonymous reviewers for their thoughtful feedback and suggestions. This work was funded by NIH R01 grants HG004483, GM080203, and HG013117. GPUs were provided in part by the NVIDIA Academic Hardware Grant.

## Author Contributions

A.L. led software development, experimental evaluation, and writing. I.H. conceptualized the project, provided reference implementations of basic pair HMMs for code testing, supervised the work, and reviewed and edited the manuscript.

## Code Availability

All scripts to reproduce Pfam dataset and train models can be found at protein_evolution_icml_2026.

## Impact Statement

A greater predictive understanding of molecular evolution processes will benefit areas of biomedical science such as epidemiology, biotechnology, and population genetics. Relevant downstream inference tasks include predictive modeling of disease-causing genetic variants, or predicting the evolutionary trajectories of viruses (like SARS-CoV-2). While designed for proteins, all methods developed here are applicable to any biological sequence, including genomes, codons, RNA transcripts, and any combination thereof.

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

# A. Solutions to transition and emission models

Here, we present the subset of derivations from tkfdp.net (Large & Holmes, 2026) needed to explain the models introduced in this work. Notation may vary slightly between this appendix and the TKF-DP appendix.

For pair HMMs, the next-column conditional probability takes the form

$$\mathcal{T}_{\text{HMM}}^{Y|X}(b \mid a, X_{i+1}, t) = \mathcal{M}(b, X_{i+1}) T_{\tau(a),\tau(b)} E(b)$$

The next-column joint probability $\mathcal{T}_{\text{HMM}}^{XY}$ has the same factorization. This section defines different forms of transition matrix $T$ and emissions scoring function $E$.

## A.1. General time reversible model, and F81 as a simplification

Let $x \in \Omega$ be an ancestral symbol and $y \in \Omega$ be a descendant symbol. For the general time reversible model (Lanave et al., 1984), the substitution rate matrix, $Q$, is parameterized by a symmetric exchangeabilities matrix $\chi$ and the equilibrium distribution $\pi$. Off-diagonal elements are $q_{xy} = \chi_{xy}\pi_y$. Diagonal elements are set such that rows of $Q$ sum to zero. $Q$ is further normalized such that $-\sum_{x \in \Omega} \pi_x Q_{xx} = 1$. This calibrates time $t = 1$ to mean the time for one substitution to occur.

The F81 substitution model (Felsenstein, 1981) is a special case of the GTR model, where all exchangeabilities are set to 1.

## A.2. Emissions scoring function

Let $b = (x, y)$ be one alignment column, and let $\tau(b)$ return the column state type. The emissions scoring function for the next-column conditional probability returns $P(y \mid x, t)$ for match columns, and $P(y)$ at insert columns.

$$E^{Y|X}(b) = \begin{cases} \exp(Qt)_{xy} & \text{if } \tau(b) = \texttt{M} \\ \pi_y & \text{if } \tau(b) = \texttt{I} \\ 1 & \text{otherwise} \end{cases}$$

where $\exp$ denotes the matrix exponential.

The emissions scoring function for the next-column joint probability includes additional terms for the ancestral symbol. It evaluates $P(x, y \mid t)$ at match columns, $P(y)$ at insert columns, and $P(x)$ at delete columns.

$$E^{XY}(b) = \begin{cases} \pi_x \exp(Qt)_{xy} & \text{if } \tau(b) = \texttt{M} \\ \pi_y & \text{if } \tau(b) = \texttt{I} \\ \pi_x & \text{if } \tau(b) = \texttt{D} \\ 1 & \text{otherwise} \end{cases}$$

## A.3. TKF91

The finite-time transition probabilities of TKF91 come from solutions to the underlying birth-death process (Thorne et al., 1991). Let $\lambda$ and $\mu$ be the rate of inserting or deleting a mortal link, respectively. The probability $\alpha$ that a mortal link survives to time $t$, the insertion probability $\beta$, the orphaned insertion probability $\gamma$ and the stationary link-sequence extension probability $\kappa$ are

$$\begin{aligned} \alpha(\lambda, \mu, t) &= \exp(-\mu t) \\ \beta(\lambda, \mu, t) &= \frac{\lambda\left(\exp(-\lambda t) - \exp(-\mu t)\right)}{\mu \exp(-\lambda t) - \lambda \exp(-\mu t)} \\ \gamma(\lambda, \mu, t) &= 1 - \frac{\mu\beta}{\lambda(1 - \alpha)} \\ \kappa(\lambda, \mu) &= \frac{\lambda}{\mu} \end{aligned}$$

The entries of the transition matrix give the probability of transitioning from one column state type to another $P(\tau(b) \mid$

$\tau(a), t)$. For the next-column conditional probability, the TKF91 transition matrix is

$$T_{\text{TKF91}}^{Y|X} = \left(\begin{array}{c|ccccc} & \text{S} & \text{M} & \text{I} & \text{D} & \text{E} \\ \hline \text{S} & 0 & (1-\beta)\alpha & \beta & (1-\beta)(1-\alpha) & (1-\beta) \\ \text{M} & 0 & (1-\beta)\alpha & \beta & (1-\beta)(1-\alpha) & (1-\beta) \\ \text{I} & 0 & (1-\beta)\alpha & \beta & (1-\beta)(1-\alpha) & (1-\beta) \\ \text{D} & 0 & (1-\gamma)\alpha & \gamma & (1-\gamma)(1-\alpha) & (1-\gamma) \\ \text{E} & 0 & 0 & 0 & 0 & 0 \end{array}\right)$$

These terms appear in future transition matrices, so we abbreviate this matrix as $V = T_{\text{TKF91}}^{Y|X}$.

For the transition matrix used in the next-column joint probability, the rate of extending the ancestor sequence (i.e. with a match or a delete state) must be factored into the appropriate transitions.

$$T_{\text{TKF91}}^{\text{XY}} = \left(\begin{array}{c|ccccc} & \text{S} & \text{M} & \text{I} & \text{D} & \text{E} \\ \hline \text{S} & 0 & V_{\text{SM}} & V_{\text{SI}} & V_{\text{SD}} & V_{\text{SE}} \\ \text{M} & 0 & \kappa V_{\text{MM}} & V_{\text{IM}} & \kappa V_{\text{DM}} & (1-\kappa)V_{\text{ME}} \\ \text{I} & 0 & \kappa V_{\text{MI}} & V_{\text{II}} & \kappa V_{\text{DI}} & (1-\kappa)V_{\text{IE}} \\ \text{D} & 0 & \kappa V_{\text{MD}} & V_{\text{ID}} & \kappa V_{\text{DD}} & (1-\kappa)V_{\text{DE}} \\ \text{E} & 0 & 0 & 0 & 0 & 0 \end{array}\right)$$

### A.4. TKF92

For the next-column conditional probability, the TKF92 (Thorne et al., 1992) transition matrix is

$$T_{\text{TKF92}}^{Y|X} = \left(\begin{array}{c|ccccc} & \text{S} & \text{M} & \text{I} & \text{D} & \text{E} \\ \hline \text{S} & 0 & V_{\text{SM}} & V_{\text{SI}} & V_{\text{SD}} & V_{\text{SE}} \\ \text{M} & 0 & \frac{1}{\nu}\left(r+(1-r)\kappa V_{\text{MM}}\right) & (1-r)V_{\text{MI}} & \frac{1}{\nu}(1-r)\kappa V_{\text{MD}} & V_{\text{ME}} \\ \text{I} & 0 & \frac{1}{\nu}(1-r)\kappa V_{\text{IM}} & r+(1-r)V_{\text{II}} & \frac{1}{\nu}(1-r)\kappa V_{\text{ID}} & V_{\text{IE}} \\ \text{D} & 0 & \frac{1}{\nu}(1-r)\kappa V_{\text{DM}} & (1-r)V_{\text{DI}} & \frac{1}{\nu}\left(r+(1-r)\kappa V_{\text{DD}}\right) & V_{\text{DE}} \\ \text{E} & 0 & 0 & 0 & 0 & 0 \end{array}\right)$$

where $\nu = r + (1-r)\frac{\lambda}{\mu}$. This form is slightly different from (Thorne et al., 1992), as it also conditions on the ancestral sequence consisting of fragments with unknown boundaries.

For the next-column joint probability, the transition matrix is

$$T_{\text{TKF92}}^{\text{XY}} = \left(\begin{array}{c|ccccc} & \text{S} & \text{M} & \text{I} & \text{D} & \text{E} \\ \hline \text{S} & 0 & \kappa V_{\text{SM}} & V_{\text{SI}} & \kappa V_{\text{SD}} & (1-\kappa)V_{\text{SE}} \\ \text{M} & 0 & r+(1-r)\kappa V_{\text{MM}} & (1-r)V_{\text{MI}} & (1-r)\kappa V_{\text{MD}} & (1-r)(1-\kappa)V_{\text{ME}} \\ \text{I} & 0 & (1-r)\kappa V_{\text{IM}} & r+(1-r)V_{\text{II}} & (1-r)\kappa V_{\text{ID}} & (1-r)(1-\kappa)V_{\text{IE}} \\ \text{D} & 0 & (1-r)\kappa V_{\text{DM}} & (1-r)V_{\text{DI}} & r+(1-r)\kappa V_{\text{DD}} & (1-r)(1-\kappa)V_{\text{DE}} \\ \text{E} & 0 & 0 & 0 & 0 & 0 \end{array}\right)$$

### A.5. Emissions scoring function for mixture of site classes

With the mixture of site classes model, we marginalize over latent class labels $c \sim \text{Categorical}(u)$ inside the emissions scoring function.

$$E_{\text{siteMix}}^{\text{XY}}(b) = \begin{cases} \sum_c u_c \pi_{x,c} \exp(Q_c t)_{xy} & \text{if } \tau(b) = \text{M} \\ \sum_c u_c \pi_{yc} & \text{if } \tau(b) = \text{I} \\ \sum_c u_c \pi_{x,c} & \text{if } \tau(b) = \text{D} \\ 1 & \text{otherwise} \end{cases}$$

The next-column joint probability is evaluated as

$$\mathcal{T}_{\text{HMM}}^{XY}(b \mid a, X_{i+1}, t) = \mathcal{M}(b, X_{i+1}) T_{\tau(a),\tau(b)}^{XY} E_{\text{siteMix}}^{\text{XY}}(b)$$

### A.6. Transition matrix for mixture of fragment classes

The set of pair HMM states is expanded to include column states augmented with fragment class labels $f \sim \text{Categorical}(w)$. Let $b'$ and $a'$ be *states* containing the current and previous columns' state and fragment labels, respectively: $b' = (\tau(b), f)$. Start and end sentinel tokens do not have a fragment class. The new TKF92-based transition matrix is

$$
T_{\texttt{fragMix}} = \begin{cases}
r_f + (1 - r_f)\, V_{\texttt{II}}\, w_g & \text{where } a' = (m, f), b' = (n, g), m = n = \texttt{I}, f = g \\
r_f + (1 - r_f)\, \kappa\, V_{nn}\, w_g & \text{where } a' = (m, f), b' = (n, g), m = n \neq \texttt{I}, f = g \\
(1 - r_f)\, V_{m\texttt{I}}\, w_g & \text{where } a' = (m, f), b' = (n, g), (m \neq n \text{ or } f \neq g), n = \texttt{I} \\
(1 - r_f)\, \kappa\, V_{mn}\, w_g & \text{where } a' = (m, f), b' = (n, g), (m \neq n \text{ or } f \neq g), n \neq \texttt{I} \\
(1 - r_f)\, (1 - \kappa)\, V_{m\texttt{E}} & \text{where } a' = (m, f), \tau(b) = \texttt{E} \\
V_{\texttt{SI}}\, w_g & \text{where } \tau(a) = \texttt{S}, b' = (n, g), n = \texttt{I} \\
\kappa\, V_{\texttt{S}n}\, w_g & \text{where } \tau(a) = \texttt{S}, b' = (n, g), n \neq \texttt{I} \\
(1 - \kappa)\, V_{\texttt{SE}} & \text{where } \tau(a) = \texttt{S}, \tau(b) = \texttt{E}
\end{cases}
$$

The fragment and site class labels are marginalized out with the Forward algorithm.

### A.7. Transition matrix for mixture of domain classes

In the pair HMM for the joint ancestral-descendant distribution, the $\texttt{S} \rightarrow \texttt{E}$ transition weight is $(1 - \kappa)(1 - \beta)$. Thus the probability that a $\texttt{M}$ state emits no sequence is $z_t = \sum_{n \in \mathcal{C}_n} v_n (1 - \kappa_n)(1 - \beta_n)$ where $\kappa_n$ and $\beta_n$ refer to birth-death parameters $\kappa$ and $\beta$ calculated from indel rates $\lambda_n$ and $\mu_n$.

The single HMM for the stationary distribution over sequences represents a special case of this where $t = 0$. In this HMM, the start$\rightarrow$end transition weight is $1 - \kappa$. Thus the probability that an $\texttt{I}$ or $\texttt{D}$ state generates a zero-length sequence is $\sum_{n \in \mathcal{C}_n} v_n (1 - \kappa_n) \equiv z_0$.

Start with the TKF92 pair HMM transition matrix and split the $\texttt{M}$, $\texttt{I}$, and $\texttt{D}$ states into non-emitting and emitting states. Lump the non-emitting $\texttt{M}$ and $\texttt{I}$ together as state $\texttt{A}$ and let the non-emitting $\texttt{D}$ be $\texttt{B}$.

This derivation builds off transition terms in the joint TKF91 transition matrix. For ease of notation, let $K = T_{\texttt{TKF91}}^{\text{XY}}$.

The $7 \times 7$ transition matrix for the exploded joint pair HMM is

$$
v = \left(
\begin{array}{c|ccccccc}
 & \texttt{S} & \texttt{M} & \texttt{I} & \texttt{D} & \texttt{E} & \texttt{A} & \texttt{B} \\
\hline
\texttt{S} & 0 & (1 - z_t)K_{\texttt{SM}} & (1 - z_0)K_{\texttt{SI}} & (1 - z_0)K_{\texttt{SD}} & K_{\texttt{SE}} & z_t K_{\texttt{SM}} + z_0 K_{\texttt{SI}} & z_0 K_{\texttt{SD}} \\
\texttt{M} & 0 & (1 - z_t)K_{\texttt{MM}} & (1 - z_0)K_{\texttt{MI}} & (1 - z_0)K_{\texttt{MD}} & K_{\texttt{ME}} & z_t K_{\texttt{MM}} + z_0 K_{\texttt{MI}} & z_0 K_{\texttt{MD}} \\
\texttt{I} & 0 & (1 - z_t)K_{\texttt{IM}} & (1 - z_0)K_{\texttt{II}} & (1 - z_0)K_{\texttt{ID}} & K_{\texttt{IE}} & z_t K_{\texttt{IM}} + z_0 K_{\texttt{II}} & z_0 K_{\texttt{ID}} \\
\texttt{D} & 0 & (1 - z_t)K_{\texttt{DM}} & (1 - z_0)K_{\texttt{DI}} & (1 - z_0)K_{\texttt{DD}} & K_{\texttt{DE}} & z_t K_{\texttt{DM}} + z_0 K_{\texttt{DI}} & z_0 K_{\texttt{DD}} \\
\texttt{E} & 0 & 0 & 0 & 0 & 0 & 0 & 0 \\
\texttt{A} & 0 & (1 - z_t)K_{\texttt{MM}} & (1 - z_0)K_{\texttt{MI}} & (1 - z_0)K_{\texttt{MD}} & K_{\texttt{ME}} & z_t K_{\texttt{MM}} + z_0 K_{\texttt{MI}} & z_0 K_{\texttt{MD}} \\
\texttt{B} & 0 & (1 - z_t)K_{\texttt{DD}} & (1 - z_0)K_{\texttt{DI}} & (1 - z_0)K_{\texttt{DD}} & K_{\texttt{DE}} & z_t K_{\texttt{DM}} + z_0 K_{\texttt{DI}} & z_0 K_{\texttt{DD}}
\end{array}
\right)
$$

where $K \equiv K^{(0)}$ is the top-level TKF91 model with parameters $(\lambda_0, \mu_0)$. Let $U_{\Phi_1, \Phi_2}$ be the matrix formed from rows $m \in \Phi_1$ and columns $n \in \Phi_2$ of $v$. Consider the matrix of transitions between the empty states $\texttt{A}, \texttt{B}$

$$
U_{\texttt{AB},\texttt{AB}} = \left(
\begin{array}{c|cc}
 & \texttt{A} & \texttt{B} \\
\hline
\texttt{A} & z_t K_{\texttt{MM}} + z_0 K_{\texttt{MI}} & z_0 K_{\texttt{MD}} \\
\texttt{B} & z_t K_{\texttt{DM}} + z_0 K_{\texttt{DI}} & z_0 K_{\texttt{DD}}
\end{array}
\right)
$$

Summing over empty paths of all lengths (including zero-length) involving $\mathtt{A}, \mathtt{B}$

$$\sum_{k=0}^{\infty} U_{\mathtt{AB},\mathtt{AB}}^k = (I - U_{\mathtt{AB},\mathtt{AB}})^{-1}$$

$$= \frac{1}{\det(I - U_{\mathtt{AB},\mathtt{AB}})} \left( \begin{array}{c|cc} & \mathtt{A} & \mathtt{B} \\ \hline \mathtt{A} & 1 - z_0 K_{\mathtt{DD}} & z_0 K_{\mathtt{MD}} \\ \mathtt{B} & z_t K_{\mathtt{DM}} + z_0 K_{\mathtt{DI}} & 1 - z_t K_{\mathtt{MM}} - z_0 K_{\mathtt{MI}} \end{array} \right)$$

$$\det(I - U_{\mathtt{AB},\mathtt{AB}}) = (1 - z_0 K_{\mathtt{DD}})(1 - z_t K_{\mathtt{MM}} - z_0 K_{\mathtt{MI}}) - (z_t K_{\mathtt{DM}} + z_0 K_{\mathtt{DI}}) z_0 K_{\mathtt{MD}}$$

The effective nonempty $5 \times 5$ transition matrix (with $\mathtt{A}, \mathtt{B}$ summed out) is

$$T' \equiv U_{\mathtt{SMIDE},\mathtt{SMIDE}} + U_{\mathtt{SMIDE},\mathtt{AB}} \cdot (I - U_{\mathtt{AB},\mathtt{AB}})^{-1} \cdot U_{\mathtt{AB},\mathtt{SMIDE}}$$

The hierarchically nested pair HMM has $2 + 5 \, |\mathcal{C}_n| \, |\mathcal{C}_f|$ states: $\{\mathtt{SS}, \mathtt{EE}\} \cup \{\mathtt{UX}_{cf} : \mathtt{UX} \in \{\mathtt{MM}, \mathtt{MI}, \mathtt{MD}, \mathtt{II}, \mathtt{DD}\}, c \in \mathcal{C}_n, f \in \mathcal{C}_f\}$

The transition matrix $T_{\mathtt{MixDom}}$ for the mixture of domain classes model has entries

| Source $i$ $(\mathtt{UX}_{cf})$ | Destination $j$ $(\mathtt{VY}_{dg})$ | Terms in transition weight $T_{ij}^{\mathrm{MixDom}} = K_{\mathtt{XE}} \times$ | $T'_{\mathtt{UV}}$ | $\times K_{\mathtt{SY}}$ | $+ \, \delta_{\mathtt{UV}}\delta_{cd}(\ldots + \delta_{\mathtt{XY}}\delta_{fg}(\ldots))$ |
|---|---|---|---|---|---|
| $\mathtt{SS}$ | $\mathtt{MY}_{dg}$ | | $T'_{\mathtt{SM}}$ | $(1 - z_t)^{-1} v_d K_{\mathtt{SY}}^{(d)} w_{dg}$ | |
| | $\mathtt{II}_{dg}$ | | $T'_{\mathtt{SI}}$ | $(1 - z_0)^{-1} v_d \kappa_d w_{dg}$ | |
| | $\mathtt{DD}_{dg}$ | | $T'_{\mathtt{SD}}$ | $(1 - z_0)^{-1} v_d \kappa_d w_{dg}$ | |
| | $\mathtt{EE}$ | | $T'_{\mathtt{SE}}$ | | |
| $\mathtt{MX}_{cf}$ | $\mathtt{MY}_{dg}$ | $(1 - r_f) K_{\mathtt{XE}}^{(c)}$ | $T'_{\mathtt{MM}}$ | $(1 - z_t)^{-1} v_d K_{\mathtt{SY}}^{(d)} w_{dg}$ | $+ \, \delta_{cd}\left( (1 - r_f) K_{\mathtt{XY}}^{(d)} w_{dg} + \delta_{\mathtt{XY}} \delta_{fg} r_g \right)$ |
| | $\mathtt{II}_{dg}$ | $(1 - r_f) K_{\mathtt{XE}}^{(c)}$ | $T'_{\mathtt{MI}}$ | $(1 - z_0)^{-1} v_d \kappa_d w_{dg}$ | |
| | $\mathtt{DD}_{dg}$ | $(1 - r_f) K_{\mathtt{XE}}^{(c)}$ | $T'_{\mathtt{MD}}$ | $(1 - z_0)^{-1} v_d \kappa_d w_{dg}$ | |
| | $\mathtt{EE}$ | $(1 - r_f) K_{\mathtt{XE}}^{(c)}$ | $T'_{\mathtt{ME}}$ | | |
| $\mathtt{II}_{cf}$ | $\mathtt{MY}_{dg}$ | $(1 - r_f)(1 - \kappa_c)$ | $T'_{\mathtt{IM}}$ | $(1 - z_t)^{-1} v_d K_{\mathtt{SY}}^{(d)} w_{dg}$ | |
| | $\mathtt{II}_{dg}$ | $(1 - r_f)(1 - \kappa_c)$ | $T'_{\mathtt{II}}$ | $(1 - z_0)^{-1} v_d \kappa_d w_{dg}$ | $+ \, \delta_{cd} \left( (1 - r_f) \kappa_d w_{dg} + \delta_{fg} r_g \right)$ |
| | $\mathtt{DD}_{dg}$ | $(1 - r_f)(1 - \kappa_c)$ | $T'_{\mathtt{ID}}$ | $(1 - z_0)^{-1} v_d \kappa_d w_{dg}$ | |
| | $\mathtt{EE}$ | $(1 - r_f)(1 - \kappa_c)$ | $T'_{\mathtt{IE}}$ | | |
| $\mathtt{DD}_{cf}$ | $\mathtt{MY}_{dg}$ | $(1 - r_f)(1 - \kappa_c)$ | $T'_{\mathtt{DM}}$ | $(1 - z_t)^{-1} v_d K_{\mathtt{SY}}^{(d)} w_{dg}$ | |
| | $\mathtt{II}_{dg}$ | $(1 - r_f)(1 - \kappa_c)$ | $T'_{\mathtt{DI}}$ | $(1 - z_0)^{-1} v_d \kappa_d w_{dg}$ | |
| | $\mathtt{DD}_{dg}$ | $(1 - r_f)(1 - \kappa_c)$ | $T'_{\mathtt{DD}}$ | $(1 - z_0)^{-1} v_d \kappa_d w_{dg}$ | $+ \, \delta_{cd} \left( (1 - r_f) \kappa_d w_{dg} + \delta_{fg} r_g \right)$ |
| | $\mathtt{EE}$ | $(1 - r_f)(1 - \kappa_c)$ | $T'_{\mathtt{DE}}$ | | |

where $K^{(n)}$ denotes the inner-level, joint TKF91 with domain class label $n$ and indel rates $(\lambda_n, \mu_n)$.

# B. Neural network architectures

### B.1. Sequence embedding architectures

Let $B(X)$ and $D(Y)$ be the neural networks that generate sequence embeddings with embedding size $H$. We will define $B(X)$ for concreteness, although the same formulae apply to $D(Y)$ (except where stated otherwise).

Both the ancestor and descendant sequences are first projected to a fixed-width vector space using learned embedding matrices, $E$,

$$h_0 = E \cdot \mathtt{onehot}(X)$$

where $E \in \mathbb{R}^{H \times |\Omega|}$.

**Residual CNN**    The Residual CNN (LeCun et al., 1989) uses convolutions to capture interactions between protein sites. This model follows a pre-layer normalization residual architecture with SiLU activation and dropout (Xiong et al., 2020; He et al., 2016; Elfwing et al., 2017; Srivastava et al., 2014).

$$
\begin{aligned}
h_1 &= \texttt{layerNorm}(h_0; \gamma, \beta) \\
h_2 &= \texttt{conv}(h_1; w_{\texttt{conv}}, b_{\texttt{conv}}) \\
h_3 &= \texttt{silu}(h_2) \\
h_4 &= \texttt{dropout}(h_3; p_{\texttt{B}}) \\
B_{\texttt{cnn}} &= h_4 + h_0
\end{aligned}
$$

The parameters are:

- $\gamma, \beta$: scale and bias for layer normalization

- $w_{\texttt{conv}} \in \mathbb{R}^{k \times H \times H}$: convolution kernel weights, with kernel width $k$

- $b_{\texttt{conv}} \in \mathbb{R}^{H}$: convolution kernel bias

- $p_{\texttt{B}}$: dropout rate

The ancestor embedding model centers the convolution kernel on the site of interest, giving it bidirectional context. The descendant embedding model uses a causal kernel, placing the site of interest at the rightmost position so its context only includes preceding sites.

We use an embedding size of $H = 1028$ and a dropout rate of $p_B = 0.20$. The ancestor embedding model uses a convolutional kernel width of 16, while the descendant model uses a kernel width of 8.

**LSTM**    This refers to an RNN that uses LSTM units, as implemented by Flax function `flax.linen.OptimizedLSTMCell` (Hochreiter & Schmidhuber, 1997; Heek et al., 2024). The ancestor embedding model uses a bidirectional RNN, with forwards and backwards embeddings concatenated.

$$
\begin{aligned}
h_1 &= \texttt{LSTM}(h_0) \\
h_2 &= \texttt{reverse}\left(\texttt{LSTM}\left(\texttt{reverse}(h_0)\right)\right) \\
B_{\texttt{LSTM}} &= h_1 \oplus h_2
\end{aligned}
$$

The descendant embedding model stops at $h_1$. We use a unidirectional RNN for embedding descendant sequences to maintain causal context. Thus, the descendant embedding dimension will be half that of the ancestor embedding dimension.

Again, we use an embedding size of $H = 1028$ and a dropout rate of $p = 0.20$.

**Transformer with Rotary Positional Encoding**    Here, we use a pre-norm Transformer block with standard dot-product self-attention and rotary positional encoding (Vaswani et al., 2017; Su et al., 2024), followed by a multilayer perceptron (MLP).

The self-attention block is

$$
\begin{aligned}
h_1 &= \texttt{layerNorm}(h_0; \gamma_1, \beta_1) \\
Q', K', V &= \texttt{linear}(h_1; w_{\texttt{Q,K,V}}, b_{\texttt{Q,K,V}}) \\
Q, K &= \texttt{RoPE}(Q', K') \\
h_2 &= \texttt{selfattention}(Q, K, V; w_{\texttt{O}}, b_{\texttt{O}}) \\
h_3 &= h_0 + \texttt{dropout}(h_2; p_B)
\end{aligned}
$$

Transformer parameters include $w_{\mathtt{Q,K,V}} \in \mathbb{R}^{H \times 3H}$, $w_{\mathtt{O}} \in \mathbb{R}^{H \times H}$, $b_{\mathtt{Q,K,V}} \in \mathbb{R}^{3H}$, and $b_{\mathtt{O}} \in \mathbb{R}^H$. The other parameters are the previously-described $\gamma_1$, $\beta_1$, and $p_{\mathtt{B}}$.

The MLP block is

$$h_4 = \mathtt{layerNorm}(h_3; \gamma_2, \beta_2)$$
$$h_5 = \mathtt{linear}(h_4; w_1, b_1)$$
$$h_6 = \mathtt{silu}(h_5)$$
$$h_7 = \mathtt{linear}(h_6; w_2, b_2)$$
$$B_{\mathtt{trans}} = h_3 + \mathtt{dropout}(h_7; p_B)$$

The parameters are $w_1, w_2 \in \mathbb{R}^{H \times H}$, biases $b_1, b_2 \in \mathbb{R}^H$, unique layer normalization parameters, $\gamma_2$ and $\beta_2$, and the same dropout rate as in the attention block, $p_{\mathtt{B}}$.

For the ancestor embedding $B_{\mathtt{trans}}$, the self-attention uses a padding mask. For the descendant embedding $D_{\mathtt{trans}}$, both a padding mask and a causal mask are applied.

For the 1-block Transformer, we use embedding size $H = 1,452$, dropout of $p_B = 0.20$, and two attention heads. For the final 6-block Transformer, we use a smaller embedding size $H = 756$ and 6 attention heads.

## B.2. Architecture of Basic neural prediction head

The column-specific embeddings are created by concatenating the sequence embeddings, previous column's state, and evolutionary time. The embeddings will have size $H'$.

$$h_0' = \mathtt{layerNorm}\left(B(X)_{i+1} \oplus D(Y)_j; \gamma_F, \beta_F\right) \oplus \mathtt{onehot}(\tau(a)) \oplus [t]$$

where $h_0' \in \mathbb{R}^{H'}$.

The prediction head, $F$, takes $h_0'$ as input and outputs logits used to calculate probabilities over the alignment-augmented alphabet, $\mathcal{T}_{\mathtt{neu}}^{Y|X}(b \mid a, i, j, X, Y_{\ldots j})$

$$h_1' = \mathtt{linear}(h_0'; w_{\mathtt{F1}}, b_{\mathtt{F1}})$$
$$h_2' = \mathtt{silu}(h_1')$$
$$h_3' = \mathtt{dropout}(h_2'; p_{\mathtt{F}})$$
$$z_{ij} = \mathtt{linear}\left(h_3'; w_{\mathtt{F2}}, b_{\mathtt{F2}}\right)$$
$$\mathcal{T}_{\mathtt{neu}}^{Y|X}(b \mid a, i, j, X, Y_{\ldots j}) = (\mathcal{U}(X_{i+1}) \cdot \mathtt{softmax}(z_{ij}))_b$$

Parameters include dropout rates and layer normalization parameters specific to the prediction head, $p_{\mathtt{F}}$, $\gamma_F$, and $\beta_F$. The intermediate weights, $w_{\mathtt{F1}} \in \mathbb{R}^{H' \times H_{\mathtt{int}}}$, project the concatenated features down to a smaller, intermediate size $H_{\mathtt{int}}$. The bias for the intermediate layer is $b_{\mathtt{F1}} \in \mathbb{R}^{H_{\mathtt{int}}}$. The final projection weight, $w_{\mathtt{F2}} \in \mathbb{R}^{H_{\mathtt{int}} \times |\Omega_{\mathtt{aug}}|}$, and bias, $b_{\mathtt{F2}} \in \mathbb{R}^{|\Omega_{\mathtt{aug}}|}$, transform the features into logits for each tuple and token in $\Omega_{\mathtt{aug}}$.

We use an internal embedding size $H_{\mathtt{int}} = 500$ and dropout rate $p_F = 0.20$

## B.3. Architecture of Neural TKF prediction head

The site- and sample-specific TKF92+F81 parameters to derive are: $\pi_{ij}, \lambda_{ij}, \mu_{ij}$, and $r_{ij}$. Instead of calculating $\lambda_{ij}$ directly, we use an offset $o_{ij}$ such that $\lambda_{ij} = \mu_{ij}(1 - o_{ij})$. Constraining $0 \le o < 1$ ensures $\mu > \lambda$.

As before, we use a small feedforward network to post-process features after concatenation. We now abbreviate this as

$$\mathtt{FF}(x; w, b, p) = \mathtt{dropout}\left(\mathtt{silu}\left(\mathtt{linear}\left(x; w, b\right)\right); p\right)$$

Empirically, we find that site-specific rates can explode to extreme values during training. To account for this, we use a modified version of the sigmoid function,

$$\texttt{sigbound}(x, b_{\texttt{min}}, b_{\texttt{max}}) = b_{\texttt{min}} + \frac{b_{\texttt{max}} - b_{\texttt{min}}}{1 + \exp(-x)}$$

**Network $W$**   produces logits for the equilibrium distribution, using the following feedforward network.

$$h'_{\pi,0} = \texttt{layerNorm}\left(B(X)_{i+1} \oplus D(Y)_j; \gamma_\pi, \beta_\pi\right)$$
$$h'_{\pi,1} = \texttt{FF}(h'_{\pi,0}; w_{\pi 1}, b_{\pi 1}, p_G)$$
$$z^\pi_{ij} = \texttt{linear}\left(h'_{\pi,1}; w_{\pi 2}, b_{\pi 2}\right)$$

where $\gamma_\pi, \beta_\pi$ are unique scales and biases for layer normalization, $p_G$ is the dropout rate for the entire prediction head. Let the size of the embeddings after concatenation be $H'_e$. Then linear weights have sizes $w_{\pi 1} \in \mathbb{R}^{H'_e \times H_{\texttt{int}}}$, $w_{\pi 2} \in \mathbb{R}^{H_{\texttt{int}} \times |\Omega|}$. Bias vectors have sizes $b_{\pi 1} \in \mathbb{R}^{H_{\texttt{int}}}$ and $b_{\pi 2} \in \mathbb{R}^{|\Omega|}$.

The equilibrium distribution is obtained by applying the softmax function

$$\pi_{ij} = \texttt{softmax}\left(z^\pi_{ij}\right)$$

$\texttt{sigbound}$ is used to produce site-specific substitution rates with the same feedforward network architecture but distinct parameters.

**Network $G$**   produces logits for evolutionary model parameters associated with the indel process. It uses a similar set of layers as $W$, except column-specific embeddings also include the previous column's state, $\tau(a)$.

$$h'_0 = \texttt{layerNorm}\left(B(X)_{i+1} \oplus D(Y)_j; \gamma, \beta\right) \oplus \texttt{onehot}(\tau(a))$$
$$h'_1 = \texttt{FF}(h'_0; w_1, b_1, p_G)$$
$$z^o_{ij} = \texttt{linear}\left(h'_1; w_{\texttt{o}}, b_{\texttt{o}}\right)$$
$$z^\mu_{ij} = \texttt{linear}\left(h'_1; w_\mu, b_\mu\right)$$
$$z^r_{ij} = \texttt{linear}\left(h'_1; w_{\texttt{r}}, b_{\texttt{r}}\right)$$

where $\gamma$ and $\beta$ are unique scales and biases for layer normalization (for a total of three unique layer normalizations across the entire prediction head). With the new embedding size after concatenation, $H'_t$, the first set of linear weights and biases have sizes $w_1 \in \mathbb{R}^{H'_t \times H_{\texttt{int}}}$, $b_1 \in \mathbb{R}^{H_{\texttt{int}}}$. The final projection weights are sizes $w_{\texttt{o}}, w_\mu, w_{\texttt{r}} \in \mathbb{R}^{H_{\texttt{int}} \times 1}$, and the final biases $b_{\texttt{o}}, b_\mu, b_{\texttt{r}}$ are scalars.

For both $G$ and $W$, we use an internal embedding size $H_{\texttt{int}} = 500$ and dropout rate $p_G = 0.20$

Indel rates are obtained with the $\texttt{sigbound}$ function, which constrains them to a fixed range.

$$o_{ij} = \texttt{sigbound}\left(z^o_{ij}, o_{\texttt{min}}, o_{\texttt{max}}\right)$$
$$\mu_{ij} = \texttt{sigbound}\left(z^\mu_{ij}, \mu_{\texttt{min}}, \mu_{\texttt{max}}\right)$$

We use limits $o_{\texttt{min}} = 10^{-4}$, $o_{\texttt{max}} = 0.333$, $\mu_{\texttt{min}} = 10^{-4}$, and $\mu_{\texttt{max}} = 2$. Subsequently, $6 \times 10^{-4} \leq \lambda_{ij} \leq 1.9998$.

The TKF92 fragment length parameter is obtained with the standard sigmoid function.

$$r_{ij} = \texttt{sigmoid}\left(z^r_{ij}\right)$$

The emission function for the neural TKF model is a site-specific F81 model

$$
E^{ij}(b) = \begin{cases} \exp(\rho_{ij}Q_{ij}t)_{xy} & \text{if } b = (x,y) \text{ where } x,y \in \Omega \\ (\pi_{ij})_y & \text{if } b = (\epsilon,y) \text{ where } y \in \Omega \\ 1 & \text{if } b = (x,\epsilon) \text{ where } x \in \Omega \\ 1 & \text{if } b = \texttt{E} \end{cases}
$$

The transition matrix is also site-specific; it depends on the ancestral and descendant sequence indices $(i,j)$

$$
T^{ij} = \begin{pmatrix}
 & \texttt{M} & \texttt{I} & \texttt{D} & \texttt{E} \\
\texttt{S} & (1-\beta_{ij})\alpha_{ij} & \beta_{ij} & (1-\beta_{ij})(1-\alpha_{ij}) & 1-\beta_{ij} \\
\texttt{M} & \frac{1}{\nu_{ij}}\left(r_{ij}+(1-r_{ij})\kappa_{ij}(1-\beta_{ij})\alpha_{ij}\right) & (1-r_{ij})\beta_{ij} & \frac{1}{\nu_{ij}}(1-r_{ij})\kappa_{ij}(1-\beta_{ij})(1-\alpha_{ij}) & 1-\beta_{ij} \\
\texttt{I} & \frac{1}{\nu_{ij}}(1-r_{ij})\kappa_{ij}(1-\beta_{ij})\alpha_{ij} & r_{ij}+(1-r_{ij})\beta_{ij} & \frac{1}{\nu_{ij}}(1-r_{ij})\kappa_{ij}(1-\beta_{ij})(1-\alpha_{ij}) & 1-\beta_{ij} \\
\texttt{D} & \frac{1}{\nu_{ij}}(1-r_{ij})\kappa_{ij}(1-\gamma_{ij})\alpha_{ij} & (1-r_{ij})\gamma_{ij} & \frac{1}{\nu_{ij}}\left(r_{ij}+(1-r_{ij})\kappa_{ij}(1-\gamma_{ij})(1-\alpha_{ij})\right) & 1-\gamma_{ij}
\end{pmatrix}
$$

Note that transitions with zero probability have been removed from this matrix.

### B.4. Hyperparameter sweeps to optimize the Transformer sequence embedding models

We perform hyperparameter sweeps using the Transformer sequence embedders and the Neural TKF prediction head. This includes repeating the sequence embedding functions (with the number of repeats referred to as number of Transformer blocks) and repeating the feedforward network $\texttt{FF}(\ldots)$ in the prediction head. The chosen values are marked with a star $(\star)$.

| Parameter | Values tested |
|---|---|
| Weight decay | $0.0, 10^{-4}, 10^{-3\star}$ |
| Learning rate | constant $10^{-4\star}$, cosine decay with linear warm-up (start at $10^{-4}$, peak at $10^{-3}$, end at $10^{-5}$) |
| Embedding size | $1452, 1028, 756^{\star}, 500, 625$ |
| Sequence embedding dropout rate, $p_B$ | $0.0, 0.2^{\star}, 0.3$ |
| Number of Transformer blocks | $1, 2, 4, 6^{\star}, 10$ |
| Number of attention heads | $1, 2, 4, 6^{\star}, 10, 12$ |
| Prediction head dropout rate, $p_G$ | $0.0^{\star}, 0.1, 0.2$ |
| Number of times $\texttt{FF}(\ldots)$ is repeated in the prediction head | $1^{\star}, 5$ |
| Feedforward intermediate embedding dimension, in prediction head | $1500, 1000, 750, 500^{\star}, 250$ |

## C. Model training

### C.1. Pair HMM-based models.

We train all pair HMM-based models with the Adam optimizer. Simple indel models (TKF91, TKF92, LG05, RS07, and H20) and mixtures of site classes are trained from matrices of total transitions and emission counts per alignment. Thus, the computational complexity of evaluating likelihoods is independent of alignment length. Mixtures of fragment and domain classes are trained using the Forward algorithm. The complexity is linear in alignment length.

We train basic indel models and the mixture of site classes for a maximum of 250 epochs, and mixture of fragment and domain class models for a maximum of 150 epochs. A counter for early stopping increments each epoch where either: 1.) negative log-likelihood (NLL) of the test set did not improve over the previous epoch by more than $1 \times 10^{-3}$ (i.e. loss is stagnant), or 2.) NLL of the test set exceeded the all-time best NLL by more than 3 (i.e. loss is worse). The counter is reset to 0 on any epoch where neither condition holds. Early stopping is triggered after 5 consecutive epochs where the counter increments.

In practice, early stopping was used for computational convenience rather than active model selection. Simple indel models optimized fewer than 15 parameters and were unlikely to have the capacity to overfit. For hierarchical mixtures, train and test loss converged to stable values well before training ended. There was always a small, consistent gap, indicating no overfitting.

Onset of the plateau in train and test set losses is determined by post-hoc analyses. Specifically, if the rolling mean of per-epoch **relative** test set loss improvement dropped below $5 \times 10^{-5}$ for 10 consecutive epochs, then the loss plateau was recorded. These analyses determine the "Loss plateau" columns for Tables S2, S4.

Most pair HMMs were trained on a single NVIDIA GeForce RTX 2080 Ti GPU (with 11GB of memory). Five of the larger mixture models were trained on a single NVIDIA A6000 GPU (with 48 GB of memory). Tables S2 - S4 give average and total times from training with "Partition 1", which is **one** train/test partition (as opposed to averaging over three partitions, as done in Table 2 of the main results).

*Table S1.* **Average, minimum, and maximum time per epoch — Indel models:** For each model trained using Partition 1, time (in seconds) between epochs was collected and aggregated across all epochs. "Total epochs" refers to the total number of epochs used to train each model. All basic pair HMMs used a batch size of 10,000 and were trained on the RTX 2080 Ti GPU.

| INDEL MODEL | TOTAL EPOCHS | AVERAGE (S/EPOCH) | MINIMUM (S/EPOCH) | MAXIMUM (S/EPOCH) |
|---|---|---|---|---|
| TKF92 | 47 | 8.85 | 8 | 9 |
| H20 | 70 | 9.74 | 9 | 11 |
| LG05 | 54 | 8.75 | 8 | 9 |
| RS07 | 49 | 8.75 | 8 | 9 |
| TKF91 | 88 | 8.79 | 8 | 9 |

*Table S2.* **Total training times — Indel models:** All models were trained using Partition 1, and training times were measured in seconds. "Best epoch" refers to the 0-indexed epoch where the best model checkpoint was created. "Loss plateau onset" and "Time to plateau" refer to the epoch where loss stabilized. After this point, only minor fluctuations in both training and test set losses were observed. All other columns are as previously described.

| INDEL MODEL | TOTAL EPOCHS | BEST EPOCH | TIME TO BEST EPOCH (S) | LOSS PLATEAU ONSET (EPOCH) | TIME TO PLATEAU (S) |
|---|---|---|---|---|---|
| TKF92 | 47 | 46 | 407 | 29 | 256 |
| H20 | 70 | 69 | 672 | 50 | 489 |
| LG05 | 54 | 53 | 464 | 34 | 298 |
| RS07 | 49 | 48 | 420 | 33 | 289 |
| TKF91 | 88 | 87 | 765 | 29 | 255 |

*Table S3.* **Average, minimum, and maximum time per epoch — Hierarchical mixtures:** For each model trained using Partition 1, time (in seconds) between epochs was collected and aggregated across all epochs. "GPU for training" refers to the single GPU used to train the model, along with its size. "Batch size" is the number of alignments per batch. "Comps." is short for "components" and refers to the number of mixture components across all levels. For example, "MixDom, 2 comps." used 2 domain classes, 2 fragment classes, and 2 site classes. All other columns are as previously described.

| Model | GPU for training | Batch size | Total epochs | Average (s/epoch) | Minimum (s/epoch) | Maximum (s/epoch) |
|---|---|---|---|---|---|---|
| MixSites, 2 comps. | RTX 2080 Ti (11GB) | 10000 | 102 | 18.95 | 18 | 20 |
| MixSites, 3 comps. | RTX 2080 Ti (11GB) | 10000 | 187 | 22.91 | 22 | 25 |
| MixSites, 4 comps. | RTX 2080 Ti (11GB) | 10000 | 250 | 24.94 | 24 | 27 |
| MixSites, 5 comps. | RTX 2080 Ti (11GB) | 10000 | 250 | 26.6 | 26 | 28 |
| MixSites, 10 comps. | RTX 2080 Ti (11GB) | 4096 | 59 | 33.41 | 33 | 35 |
| MixSites, 20 comps. | RTX 2080 Ti (11GB) | 4096 | 103 | 87.14 | 86 | 90 |
| MixSites, 30 comps. | RTX 2080 Ti (11GB) | 8000 | 147 | 127.84 | 127 | 129 |
| MixSites, 175 comps. | A6000 (48GB) | 5000 | 247 | 157.63 | 156 | 158 |
| MixSites, 500 comps. | A6000 (48GB) | 1000 | 97 | 382.96 | 381 | 384 |
| MixSites, 900 comps. | A6000 (48GB) | 1000 | 121 | 677.04 | 676 | 678 |
| MixFrag, 2 comps. | RTX 2080 Ti (11GB) | 2500 | 82 | 40.57 | 40 | 42 |
| MixFrag, 3 comps. | RTX 2080 Ti (11GB) | 2500 | 47 | 56.48 | 55 | 59 |
| MixFrag, 4 comps. | RTX 2080 Ti (11GB) | 2500 | 64 | 74.25 | 73 | 76 |
| MixFrag, 5 comps. | RTX 2080 Ti (11GB) | 2500 | 150 | 98.85 | 98 | 101 |
| MixFrag, 10 comps. | RTX 2080 Ti (11GB) | 16 | 124 | 1517.04 | 1502 | 1536 |
| MixFrag, 20 comps. | RTX 2080 Ti (11GB) | 64 | 150 | 988.56 | 984 | 997 |
| MixFrag, 30 comps. | RTX 2080 Ti (11GB) | 64 | 150 | 1730.51 | 1713 | 1755 |
| MixDom, 2 comps. | RTX 2080 Ti (11GB) | 10000 | 148 | 67.52 | 67 | 75 |
| MixDom, 3 comps. | A6000 (48GB) | 6500 | 150 | 106.01 | 105 | 111 |
| MixDom, 4 comps. | RTX 2080 Ti (11GB) | 2056 | 122 | 414.79 | 413 | 422 |
| MixDom, 5 comps. | RTX 2080 Ti (11GB) | 1028 | 103 | 889.6 | 885 | 897 |
| MixDom, 10 comps. | A6000 (48GB) | 512 | 127 | 6683.48 | 6670 | 6697 |

*Table S4.* **Total training times — Hierarchical mixtures:** All models were trained using Partition 1, and training times were measured in seconds. All other columns are as previously described.

| MODEL | GPU FOR TRAINING | BATCH SIZE | TOTAL EPOCHS | BEST EPOCH | TIME TO BEST EPOCH (S) | LOSS PLATEAU ONSET (EPOCH) | TIME TO PLATEAU (S) |
|---|---|---|---|---|---|---|---|
| MIXSITES, 2 COMPS. | RTX 2080 TI (11GB) | 10000 | 102 | 94 | 1782 | 26 | 492 |
| MIXSITES, 3 COMPS. | RTX 2080 TI (11GB) | 10000 | 187 | 136 | 3114 | 26 | 609 |
| MIXSITES, 4 COMPS. | RTX 2080 TI (11GB) | 10000 | 250 | 245 | 6111 | 33 | 824 |
| MIXSITES, 5 COMPS. | RTX 2080 TI (11GB) | 10000 | 250 | 227 | 6038 | 27 | 720 |
| MIXSITES, 10 COMPS. | RTX 2080 TI (11GB) | 4096 | 59 | 53 | 1771 | 17 | 569 |
| MIXSITES, 20 COMPS. | RTX 2080 TI (11GB) | 4096 | 103 | 97 | 8451 | 17 | 1480 |
| MIXSITES, 30 COMPS. | RTX 2080 TI (11GB) | 8000 | 147 | 142 | 18152 | 30 | 3834 |
| MIXSITES, 175 COMPS. | A6000 (48GB) | 5000 | 247 | 228 | 35940 | 16 | 2521 |
| MIXSITES, 500 COMPS. | A6000 (48GB) | 1000 | 97 | 91 | 34850 | 5 | 1916 |
| MIXSITES, 900 COMPS. | A6000 (48GB) | 1000 | 121 | 113 | 76507 | 5 | 3384 |
| MIXFRAG, 2 COMPS. | RTX 2080 TI (11GB) | 2500 | 82 | 45 | 1822 | 10 | 404 |
| MIXFRAG, 3 COMPS. | RTX 2080 TI (11GB) | 2500 | 47 | 41 | 2315 | 12 | 672 |
| MIXFRAG, 4 COMPS. | RTX 2080 TI (11GB) | 2500 | 64 | 56 | 4160 | 13 | 965 |
| MIXFRAG, 5 COMPS. | RTX 2080 TI (11GB) | 2500 | 150 | 104 | 10283 | 14 | 1388 |
| MIXFRAG, 10 COMPS. | RTX 2080 TI (11GB) | 16 | 124 | 93 | 141104 | 11 | 16686 |
| MIXFRAG, 20 COMPS. | RTX 2080 TI (11GB) | 64 | 150 | 77 | 76144 | 3 | 2967 |
| MIXFRAG, 30 COMPS. | RTX 2080 TI (11GB) | 64 | 150 | 147 | 254399 | 4 | 6977 |
| MIXDOM, 2 COMPS. | RTX 2080 TI (11GB) | 10000 | 148 | 77 | 5204 | 47 | 3184 |
| MIXDOM, 3 COMPS. | A6000 (48GB) | 6500 | 150 | 124 | 13143 | 41 | 4353 |
| MIXDOM, 4 COMPS. | RTX 2080 TI (11GB) | 2056 | 122 | 111 | 46044 | 25 | 10386 |
| MIXDOM, 5 COMPS. | RTX 2080 TI (11GB) | 1028 | 103 | 98 | 87184 | 27 | 24021 |
| MIXDOM, 10 COMPS. | A6000 (48GB) | 512 | 127 | 121 | 808746 | 10 | 66871 |

### C.2. Neural networks.

We train all neural models with the Adam optimizer. A counter for early stopping increments each epoch where either: 1.) negative log-likelihood (NLL) of the dev set did not improve over the previous epoch by more than $1 \times 10^{-3}$ (i.e. loss is stagnant), or 2.) NLL of the dev set exceeded the all-time best NLL by more than $3$ (i.e. loss is worse). The counter is reset to $0$ on any epoch where neither condition holds. Early stopping is triggered after $5$ consecutive epochs where the counter increments.

We train models with CNN sequence embedders on a single RTX 2080 Ti GPU (11GB). We train all other neural models on a single A6000 GPU (48GB). Tables S5 and S6 give average and total times from training with "Partition 2", which is a distinct train-dev-test partition from the previously mentioned Partition 1. Timing data from Partition 1 was incomplete due to training interruptions.

*Table S5.* **Average, minimum, and maximum time per epoch — Neural networks:** For each model trained using Partition 2, time (in seconds) between epochs was collected and aggregated across all epochs. "Transf." is short for Transformer. All columns are as previously described.

| MODEL | GPU FOR TRAINING | BATCH SIZE | TOTAL EPOCHS | AVERAGE (S/EPOCH) | MINIMUM (S/EPOCH) | MAXIMUM (S/EPOCH) |
|---|---|---|---|---|---|---|
| NEURAL TKF, RESIDUAL CNN | RTX 2080 TI (11GB) | 64 | 25 | 6782.04 | 6756 | 6816 |
| NEURAL TKF, LSTM | A6000 (48GB) | 128 | 9 | 6816.25 | 6796 | 6909 |
| NEURAL TKF, 1-BLOCK TRANSF. | A6000 (48GB) | 64 | 30 | 2944.62 | 2939 | 2955 |
| NEURAL TKF, 6-BLOCK TRANSF. | A6000 (48GB) | 50 | 16 | 6745.4 | 6742 | 6749 |
| BASIC NEURAL, RESIDUAL CNN | RTX 2080 TI (11GB) | 64 | 30 | 6207.41 | 6107 | 6290 |
| BASIC NEURAL, LSTM | A6000 (48GB) | 128 | 14 | 6481.62 | 6474 | 6494 |
| BASIC NEURAL, 1-BLOCK TRANSF. | A6000 (48GB) | 64 | 30 | 2648.0 | 2617 | 2659 |
| BASIC NEURAL, 6-BLOCK TRANSF. | A6000 (48GB) | 50 | 14 | 6614.92 | 6567 | 6638 |

*Table S6.* **Total training times — Neural networks:** All models were trained using Partition 2, and training times were measured in seconds. "Final epoch" refers to the 0-indexed epoch when early stopping activated. All other columns are as previously described.

| MODEL | GPU FOR TRAINING | BATCH SIZE | TOTAL EPOCHS | TIME TO FINAL EPOCH (S) | BEST EPOCH | TIME TO BEST EPOCH (S) |
|---|---|---|---|---|---|---|
| NEURAL TKF, RESIDUAL CNN | RTX 2080 TI (11GB) | 64 | 25 | 162769 | 3 | 20403 |
| NEURAL TKF, LSTM | A6000 (48GB) | 128 | 9 | 54530 | 3 | 20509 |
| NEURAL TKF, 1-BLOCK TRANSF. | A6000 (48GB) | 64 | 30 | 85394 | 29 | 85394 |
| NEURAL TKF, 6-BLOCK TRANSF. | A6000 (48GB) | 50 | 16 | 101181 | 7 | 47226 |
| BASIC NEURAL, RESIDUAL CNN | RTX 2080 TI (11GB) | 64 | 30 | 180015 | 29 | 180015 |
| BASIC NEURAL, LSTM | A6000 (48GB) | 128 | 14 | 84261 | 3 | 19446 |
| BASIC NEURAL, 1-BLOCK TRANSF. | A6000 (48GB) | 64 | 30 | 76792 | 24 | 63599 |
| BASIC NEURAL, 6-BLOCK TRANSF. | A6000 (48GB) | 50 | 14 | 85994 | 8 | 53028 |

## D. Additional results: evaluation of approximations to the GGI process

In Table S7, we repeat the indel model comparison used to build Table 1 with the GTR substitution model using LG08 exchangeabilities (GTR-LG08) (Le & Gascuel, 2008). Results yielded the same conclusions, demonstrating that TKF92 is the best indel model for further elaborations, regardless of substitution model.

*Table S7.* **Comparing indel models by NLL of the held-out test set (lower NLL and ECE are better), with GTR-LG08:** Columns are the same as Table 1. Metrics for the best model, TKF92, are **bolded**.

| INDEL MODEL | TOTAL NLL $(\times 10^6)$ | AVERAGE NLL | ECE |
|---|---|---|---|
| TKF92 | **71.58** | **280.240** | **6.1763** |
| H20 | 71.64 | 280.461 | 6.1875 |
| LG05 | 74.08 | 290.002 | 6.5310 |
| RS07 | 74.09 | 290.032 | 6.5321 |
| TKF91 | 75.70 | 296.362 | 6.7533 |

In Table S8, we repeat the indel model comparison again, except we calculate likelihoods by marginalizing over a geometric grid of branch lengths. The motivation is that branch lengths provided by Pfam are constructed under a model of molecular evolution and alignment algorithm without an explicit indel model. We define a geometric grid of 63 branch lengths centered at 0.1 with multiplicative step size 1.13 between consecutive points (i.e. values in range $[0.1 \times 1.13^{-31}, 0.1 \times 1.13^{31}]$, which is $[0.002262, 4.420]$). This grid closely approximates the range of branch lengths provided by Pfam. When we evaluate likelihoods, we assume an exponential prior over time with parameter 1.116, again based on Pfam branch lengths.

Likelihoods were largely unchanged and yielded the same conclusions as before. Since results are consistent whether we treat branch length as an unknown to marginalize out or as a known value, further experiments with more computationally intensive models used Pfam branch lengths only. While all experiments could be repeated with the same grid of times, this would increase the compute burden by 63-fold.

*Table S8.* **Comparing indel models by NLL of the held-out test set (lower NLL and ECE are better), with GTR-LG08 and geometric grid of branch lengths:** Columns are the same as Table 1. Metrics for the best model, TKF92, are **bolded**.

| INDEL MODEL | TOTAL NLL $(\times 10^6)$ | AVERAGE NLL | ECE |
|---|---|---|---|
| TKF92 | **71.84** | **281.225** | **6.2587** |
| H20 | 71.89 | 281.439 | 6.2699 |
| LG05 | 74.26 | 290.727 | 6.6065 |
| RS07 | 74.28 | 290.791 | 6.6090 |
| TKF91 | 75.79 | 296.718 | 6.8161 |

## E. Likelihoods compared to Progen 2

In Table S9, we tabulate final likelihood-based metrics for models trained on Partition 1. We also include test set likelihood metrics for Progen 2 (Nijkamp et al., 2023), an autoregressive large protein language model trained on single sequences. This directly compares $P(Y)$ (evaluated by Progen 2) against $P(Z \mid X, t)$ (evaluated by models here).

*Table S9.* **Comparing neural models against TKF-based hierarchical mixture models by NLL of held-out test set (lower NLL and ECE are better):** Likelihood-based metrics for Progen2 come from evaluating $P(Y)$, while all other models evaluate $P(Z \mid X, t)$. Best metrics are **bolded**. Refer to Table 2 for descriptions of all other columns and symbols.

| MODEL TYPE | MODEL SUBTYPE | NUMBER OF PARAMETERS | TOTAL NLL $(\times 10^6)$ | AVERAGE NLL | ECE |
|---|---|---|---|---|---|
| NEURAL TKF (SEC. 3.3) | 6-BLOCK TRANSFORMER | **43.55 M** | **60.76** | **248.282** | **5.5118** |
| NEURAL TKF | LSTM | 30.08 M | 60.93 | 249.002 | 5.5344 |
| NEURAL TKF | RESIDUAL CNN | 28.53 M | 61.60 | 251.726 | 5.6363 |
| NEURAL TKF | 1-BLOCK TRANSFORMER | 29.78 M | 62.10 | 253.778 | 5.7016 |
| BASIC NEURAL (SEC. 3.2) | LSTM | 27.00 M | 61.84 | 252.690 | 5.6915 |
| BASIC NEURAL | 6-BLOCK TRANSFORMER | 42.04 M | 62.06 | 253.594 | 5.7301 |
| BASIC NEURAL | RESIDUAL CNN | 26.47 M | 62.25 | 254.369 | 5.7489 |
| BASIC NEURAL | 1-BLOCK TRANSFORMER | 26.87 M | 63.11 | 257.894 | 5.8888 |
| MIXTURE OF
. . . DOMAIN CLASSES (SEC. 3.1.2) | 10 COMPS. | $\star$ 29.23 K | $\star$ 61.27 | $\star$ 250.392 | $\star$ 5.7041 |
| . . . FRAGMENT CLASSES (SEC. 3.1.1) | 30 COMPS. ($k_n = 1$) | 26.16 K | 62.95 | 257.229 | 5.9612 |
| . . . SITE CLASSES (SEC. 2.6) | 30 COMPS. ($k_n = k_f = 1$) | 873 | 64.02 | 261.625 | 6.1177 |
| TKF92+F81 (SEC. 2.4) | N/A | 11 | 69.32 | 283.281 | 7.1328 |
| TKF91+F81 (SEC. 2.4) | N/A | 10 | 72.75 | 297.302 | 7.7750 |
| CONDITIONAL COLUMN FREQUENCIES | N/A | 0 | 79.34 | 315.392 | 7.2107 |
| PROGEN2 | LARGE | 2.7B | 74.44 | 303.549 | 8.9328 |

## F. Stratification: Likelihoods of most distant alignments

In the following tables, we stratify results from Partition 1 (the same partition tabulated in Table S9) by descendant sequence length (Table S10), branch length (Table S11), ungapped sequence identity (Table S12), and percent indels in alignment (Table S13). Neural TKF with the 6-block Transformer sequence embedding networks continues to be the best model regardless of descendant sequence length. However, the 10-component mixture of domain classes has the best fit to more distant alignments (i.e. those with largest branch lengths, lowest percent sequence identity, and gappiest alignments), even exceeding the performance of the best Neural TKF model.

*Table S10.* **ECE, stratified by descendant sequence length (lower is better).** All models are the same as those in Table 2. Shortest 5%, 10%, and 20% refer to the respective percentage of test set alignments with the shortest descendant sequences. Longest 5%, 10%, and 20% refer to the respective percentage of test set alignments with the longest descendant sequences. "All Data" is the ECE from the test set Partition 1. Best per column in **bold**.

| MODEL | SHORTEST 5% | SHORTEST 10% | SHORTEST 20% | ALL DATA | LONGEST 20% | LONGEST 10% | LONGEST 5% |
|---|---|---|---|---|---|---|---|
| NEURAL TKF, 6-BLOCK TRANSF. | 6.2693 | 5.9017 | **5.4378** | **5.5118** | **5.2146** | **4.9471** | **4.8056** |
| NEURAL TKF, LSTM | **6.2498** | **5.8973** | 5.4405 | 5.5344 | 5.2410 | 4.9704 | 4.8271 |
| NEURAL TKF, RESIDUAL CNN | 6.3633 | 6.0020 | 5.5353 | 5.6363 | 5.3419 | 5.0606 | 4.9105 |
| NEURAL TKF, 1-BLOCK TRANSF. | 6.3254 | 5.9880 | 5.5419 | 5.7016 | 5.4093 | 5.1307 | 4.9767 |
| BASIC NEURAL, 6-BLOCK TRANSF. | 6.5769 | 6.1977 | 5.6948 | 5.7301 | 5.3964 | 5.1213 | 4.9634 |
| BASIC NEURAL, LSTM | 6.5104 | 6.1315 | 5.6287 | 5.6915 | 5.3547 | 5.0727 | 4.9236 |
| BASIC NEURAL, RESIDUAL CNN | 6.5344 | 6.1595 | 5.6680 | 5.7489 | 5.4265 | 5.1416 | 4.9900 |
| BASIC NEURAL, 1-BLOCK TRANSF. | 6.6996 | 6.3138 | 5.7997 | 5.8888 | 5.5598 | 5.2639 | 5.1007 |
| MIXDOM, 10 COMPS. | 6.8019 | 6.3560 | 5.7839 | 5.7041 | 5.3215 | 5.0457 | 4.8918 |
| MIXFRAG, 30 COMPS. | 7.0155 | 6.5610 | 5.9710 | 5.9612 | 5.5831 | 5.2762 | 5.1092 |
| MIXSITES, 30 COMPS. | 7.0778 | 6.6365 | 6.0664 | 6.1177 | 5.7833 | 5.4789 | 5.3130 |
| TKF92+F81 | 8.4120 | 7.8341 | 7.1670 | 7.1328 | 6.6720 | 6.3137 | 6.1272 |
| TKF91+F81 | 8.7535 | 8.1658 | 7.5227 | 7.7750 | 7.3918 | 6.9557 | 6.7378 |

*Table S11.* **ECE, stratified by branch length (lower is better).** Model names are the same as before. Shortest 5%, 10%, and 20% refer to the respective percentage of test set alignments with the shortest branch lengths between ancestor and descendant. Longest 5%, 10%, and 20% refer to the respective percentage of test set alignments with the longest branch lengths. Best per column in **bold**.

| MODEL | SHORTEST 5% | SHORTEST 10% | SHORTEST 20% | ALL DATA | LONGEST 20% | LONGEST 10% | LONGEST 5% |
|---|---|---|---|---|---|---|---|
| NEURAL TKF, 6-BLOCK TRANSF. | **2.3515** | **2.5198** | **2.7388** | 5.5118 | **12.9927** | 15.4963 | 17.4799 |
| NEURAL TKF, LSTM | 2.3597 | 2.5277 | 2.7471 | 5.5344 | 13.0546 | 15.5726 | 17.5746 |
| NEURAL TKF, RESIDUAL CNN | 2.3708 | 2.5409 | 2.7641 | 5.6363 | 13.5222 | 16.2125 | 18.3745 |
| NEURAL TKF, 1-BLOCK TRANSF. | 2.3845 | 2.5581 | 2.7866 | 5.7016 | 13.6028 | 16.2706 | 18.3913 |
| BASIC NEURAL, 6-BLOCK TRANSF. | 2.6147 | 2.7469 | 2.9390 | 5.7301 | 13.7007 | 16.5771 | 18.8895 |
| BASIC NEURAL, LSTM | 2.5045 | 2.6483 | 2.8480 | 5.6915 | 13.8738 | 16.8128 | 19.1817 |
| BASIC NEURAL, RESIDUAL CNN | 2.5159 | 2.6627 | 2.8662 | 5.7489 | 13.9956 | 16.9306 | 19.2894 |
| BASIC NEURAL, 1-BLOCK TRANSF. | 2.5240 | 2.6706 | 2.8779 | 5.8888 | 14.9788 | 18.3738 | 21.1555 |
| MIXDOM, 10 COMPS. | 2.3903 | 2.5741 | 2.8123 | 5.7041 | 13.0904 | **15.2365** | **16.7652** |
| MIXFRAG, 30 COMPS. | 2.4297 | 2.6124 | 2.8561 | 5.9612 | 14.5249 | 17.3615 | 19.5709 |
| MIXSITES, 30 COMPS. | 2.4655 | 2.6545 | 2.9088 | 6.1177 | 14.8709 | 17.7045 | 19.8936 |
| TKF92+F81 | 2.7907 | 3.0339 | 3.3479 | 7.1328 | 16.7500 | 19.4400 | 21.3638 |
| TKF91+F81 | 2.8947 | 3.1268 | 3.4470 | 7.7750 | 20.5812 | 25.1040 | 28.8561 |

*Table S12.* **ECE by sequence identity subgroup (lower is better).** Model names are the same as before. Lowest 5%, 10%, and 20% refer to the respective percentage of test set alignments with the lowest percent similarity between ungapped ancestor and descendant sequences. Highest 5%, 10%, and 20% refer to the respective percentage of test set alignments with the highest percent similarity between sequences. Best per column in **bold**.

| MODEL | LOWEST 5% | LOWEST 10% | LOWEST 20% | ALL DATA | HIGHEST 20% | HIGHEST 10% | HIGHEST 5% |
|---|---|---|---|---|---|---|---|
| NEURAL TKF, 6-BLOCK TRANSF. | 17.9774 | 15.9092 | **13.2256** | **5.5118** | **2.7136** | **2.4840** | **2.3126** |
| NEURAL TKF, LSTM | 18.0514 | 15.9736 | 13.2820 | 5.5344 | 2.7210 | 2.4907 | 2.3193 |
| NEURAL TKF, RESIDUAL CNN | 18.8695 | 16.6176 | 13.7515 | 5.6363 | 2.7388 | 2.5054 | 2.3324 |
| NEURAL TKF, 1-BLOCK TRANSF. | 18.8602 | 16.6592 | 13.8198 | 5.7016 | 2.7618 | 2.5242 | 2.3462 |
| BASIC NEURAL, 6-BLOCK TRANSF. | 19.5814 | 17.1202 | 13.9889 | 5.7301 | 2.9182 | 2.7167 | 2.5804 |
| BASIC NEURAL, LSTM | 19.8638 | 17.3443 | 14.1588 | 5.6915 | 2.8254 | 2.6178 | 2.4708 |
| BASIC NEURAL, RESIDUAL CNN | 19.9450 | 17.4488 | 14.2784 | 5.7489 | 2.8446 | 2.6341 | 2.4856 |
| BASIC NEURAL, 1-BLOCK TRANSF. | 21.9391 | 18.9735 | 15.2890 | 5.8888 | 2.8550 | 2.6403 | 2.4894 |
| MIXDOM, 10 COMPS. | **17.3100** | **15.6612** | 13.3414 | 5.7041 | 2.7902 | 2.5448 | 2.3599 |
| MIXFRAG, 30 COMPS. | 20.2437 | 17.8993 | 14.8330 | 5.9612 | 2.8315 | 2.5801 | 2.3945 |
| MIXSITES, 30 COMPS. | 20.5256 | 18.2064 | 15.1529 | 6.1177 | 2.8854 | 2.6242 | 2.4323 |
| TKF92+F81 | 22.2882 | 20.2238 | 17.2115 | 7.1328 | 3.2984 | 2.9670 | 2.7179 |
| TKF91+F81 | 30.1405 | 26.2199 | 21.1815 | 7.7750 | 3.3916 | 3.0503 | 2.8075 |

*Table S13.* **ECE by indel rate subgroup (lower is better).** No indels refers to the test set alignments that contained no gaps. Gappiest 5%, 10%, and 20% refer to the respective percentage of test set alignments with the most gap characters. Best per column in **bold**.

| MODEL | NO INDELS | ALL DATA | GAPPIEST 20% | GAPPIEST 10% | GAPPIEST 5% |
|---|---|---|---|---|---|
| NEURAL TKF, 6-BLOCK TRANSF. | **3.8489** | **5.5118** | 11.0915 | 13.4854 | 15.9046 |
| NEURAL TKF, LSTM | 3.8596 | 5.5344 | 11.1420 | 13.5277 | 15.9140 |
| NEURAL TKF, RESIDUAL CNN | 3.9093 | 5.6363 | 11.4557 | 13.9413 | 16.4245 |
| NEURAL TKF, 1-BLOCK TRANSF. | 3.9373 | 5.7016 | 11.6031 | 14.1556 | 16.7313 |
| BASIC NEURAL, 6-BLOCK TRANSF. | 4.0309 | 5.7301 | 11.5366 | 14.0580 | 16.5861 |
| BASIC NEURAL, LSTM | 3.9639 | 5.6915 | 11.5988 | 14.1267 | 16.6475 |
| BASIC NEURAL, RESIDUAL CNN | 3.9952 | 5.7489 | 11.7059 | 14.2537 | 16.7993 |
| BASIC NEURAL, 1-BLOCK TRANSF. | 4.0459 | 5.8888 | 12.3615 | 15.1601 | 17.9474 |
| MIXDOM, 10 COMPS. | 4.0580 | 5.7041 | **10.6750** | **12.2332** | **13.4763** |
| MIXFRAG, 30 COMPS. | 4.0838 | 5.9612 | 12.3749 | 14.9928 | 17.5000 |
| MIXSITES, 30 COMPS. | 4.1632 | 6.1177 | 12.7598 | 15.4902 | 18.1542 |
| TKF92+F81 | 4.9338 | 7.1328 | 14.3402 | 17.1852 | 19.9402 |
| TKF91+F81 | 5.0428 | 7.7750 | 19.5905 | 27.3280 | 37.3385 |

