# OpenReview forum: "Nested birth-death processes are competitive with neural networks as time-dependent models of protein evolution"
_ICML.cc/2026/Conference — ICML 2026 regular_

### Official Review · Reviewer_wWsj · 2026-03-04

**Soundness:** 3
**Presentation:** 4
**Significance:** 3
**Originality:** 4
**Overall Recommendation:** 4
**Confidence:** 3

**Summary:**

The authors explore sequence evolution models that support insertions and deletions (indels). Firstly, they extend the model from TKF92, enabling indel rates to depend on local context in the sequence, and show that this more flexible model achieves better performance on real-world sequence alignments from Pfam. Secondly, they train autoregressive deep learning models with various architectures to generate aligned sequences, achieving slightly worse performance than the best closed-form model. Thirdly, they adapt the neural architectures to instead predict the parameters of a TKF92 model, showing this significantly improves performance for the neural methods.

**Compliance With Llm Reviewing Policy:**

Affirmed.

**Final Justification:**

On account of the authors' Reply Rebuttal, I have decided to raise my score to a 4 (Weak Accept). I still feel like the main obstacle is the incohesive narrative, but the proposed changes should help with this, and there are several interesting and sound contributions that merit publcation.

**Key Questions For Authors:**

1. Why not include bigger TKF-based hierarchical mixtures in Table 2? Based on Figure 1, using more than 10 domain classes, or 10 domain classes with $k_n = k_f = 30$, seems like it could further improve performance, rendering the neural methods moot.
2. Can you provide more information on how long it took to train the neural models? For example, approximately how many GPU hours did it take for one training run of each architecture? Also, did you train the neural architectures for just a single epoch of the training data, or if not, what was the stopping criterion?
3. Can you please clarify how the parameters for the hierarchical TKF-based mixtures are esimated in practice? Some more detail on this would be useful to include in the Appendix. I am also curious how long these calculations take, compared to training a neural architecture from scratch.
1. Why does it make sense for the Neural TKF model to assume a distinct TKF92 model at every alignment column? Shouldn't there be some dependence between the parameters in different parts of a given sequence? For example, my understanding is that the parameter $r$ should be shared across the whole sequence under the TKF92 model, or at the very least be shared across a given fragment. In general, I find it slightly surprising that the Basic neural models don't perform better, and I'm trying to develop a stronger intuition for why this might be.
4. Can you please confirm why the TKF92+F81 values in Table 2 don't match TKF92 from Table 1? Is it because F81 is used instead of GTR-LG08?
5. In Appendix B.3, why do we constrain $\mu > \lambda$?

**Limitations:**

Impact statement is good but a discussion of limitations or directions for future work should be added.

**Strengths And Weaknesses:**

Strengths:

- Modeling indels and site-specific selection pressures is an important challenge in phylogenetics.
- The authors explore a variety of compelling models, including novel statistical and neural methods. This allows them to draw several interesting conclusions.
- Throughout the paper, the experiments are logical and well-justified, and claims are well-supported by evidence.
- The presentation is quite strong, with a good discussion of the background, exposition of the new models, and formatting of results.

Weaknesses:

1. There seems to be a disconnect between the evaluation procedure and over-arching goal of obtaining a better model of protein evolution. The authors evaluate likelihoods of alignments between a pair of proteins A and B. The protein pairs are obtained from the same protein family in Pfam, by iteratively selecting the two closest sequences in the phylogenetic tree for the family infered by FastTree. Each pair is considered as two samples: one with A as the ancestor and B as the descendant, and one with the roles reversed. However, in reality, neither of the sequences is actually an ancestor of the other, as they have a lowest common ancestor that is unobserved. Moreover, a strong model for sequence evolution should also be able to fit more distant proteins from the same family, rather than just the closest pairs. Thus, I feel that the experiments come short of telling us about the best model of protein evolution, as the paper's title implies.

2. The big-picture narrative of the paper is a little incohesive, since there are a few seemingly orthogonal contributions. The TKF-based hierarchical mixtures are neat, but seem mostly independent from the comparison between Basic neural and Neural TKF.

---

> ### Author Rebuttal · Authors · 2026-03-31
>
> We thank the reviewer for the constructive feedback. We address these below, with W referring to weakness and Q referring to questions.
>
> **W1:**
>
> *“Neither of the sequences is an ancestor of the other”*
>
> Because we restrict ourselves to time-reversible models, the direction of time is actually unidentifiable. By Felsenstein’s pulley principle (Felsenstein 1981), we may re-root such that either sibling can be treated as the parent. Time-reversibility is a strong assumption, but it is technically convenient and commonly used across statistical phylogenetics. We appreciate the reviewer reminding us that this limitation should be introduced more clearly.
>
> *On using cherries from the phylogenetic tree*
>
> CherryML (Prillo et al 2023) demonstrated that the likelihood of a multiple sequence alignment is reasonably approximated by decomposing the full phylogenetic tree into cherries. A key limitation is that authors did not include an indel model. To prove the same approximation holds WITH an indel model, we’d ideally compare cherry likelihoods against full tree likelihoods. Tractably evaluating full tree likelihoods with an indel model is a significant challenge in statistical phylogenetics, but one we plan to address in future work.
>
> **W2:** *“Why not include bigger TKF-based hierarchical mixtures? [...] seems like it could further improve performance, rendering the neural methods moot.”*
>
> For the camera-ready version, we can add larger hierarchical mixtures in Table 2. This may surpass our current neural network implementations. Statistical models with thousands of parameters outperforming neural methods with millions would itself be a notable result.
>
> The neural methods still have potential, regardless of outcome. Based on the training behavior of all hierarchical mixture models and the mixture of site class’s curve in Figure 1 (orange), we anticipate that these models will eventually hit a ceiling in capacity. On the other hand, the neural methods have more room to grow. We anticipate leveraging pre-trained large protein language model embeddings such as ESM (Hayes et al 2025) would greatly improve all neural models’ fit to alignments.
>
> **Q2: neural model training**
> We train all neural models with the Adam optimizer for up to 30 epochs. We employ early stopping by measuring the likelihood of a dev set. If this metric remained stationary or increased for 5 continuous epochs, then training was stopped.
>
> We train models with CNN sequence embedders on a single GeForce RTX 2080 ti (11GB). We train all other neural models on a single A6000 (48GB). Regarding length of training: no model trained for longer than 3 days. Our largest model, the neural TKF with 6-block transformers, finished training in 1 day and averaged 112.6 mins per epoch. Full tables of times and training details can be provided in the camera-ready version.
>
> **Q3: hierarchical model training**
> We train all hierarchical mixture models with the Adam optimizer. Parameter values are constrained to valid ranges with appropriate activations. We allow for a maximum of 150 epochs, and we employ early stopping based on the convergence of the test set likelihood. If this metric remained stationary for 5 continuous epochs, then training was stopped.
>
> Smaller pair HMMs are trained on the 2080 ti, while larger mixture of fragment class models and mixture of domain class models are trained on the A6000. The largest mixture of domain classes model reached convergence in about 1.5 days and averaged 109.5 min per epoch. Full tables of times and training details can be provided in the camera-ready version.
>
> **Q4:**
>
> *On using site-specific TKF92*
>
> We use a TKF92 model at every column to capture site-specific indel rates. This is like a mixture of fragment classes model, but with an infinite number of components. Similarly, SiteRM (Prillo et al 2024) assumed a substitution model at every alignment column.
>
> *On sharing parameters*
>
> For an explanation of TKF92 fragment boundaries, see our response to Reviewer 1, Tzka (Q3). Parameters cannot be associated with specific fragments because they are estimated by marginalizing over all possible fragment boundaries. For a discussion on epistasis, see our response to Reviewer 3, TLkk (W1).
>
> **Q5:** Yes, the difference between Table 1 and Table 2 is the choice of substitution model. An updated table with F81 is below.
>
> | Indel Model | Total negative log-likelihood (NLL, $\times 10^6$) | Average NLL | Exponentiated cross-entropy (ECE) |
> |---|---|---|---|
> | tkf92 | 70.3 | 281.203 | 7.1398 |
> | h20 | 70.35 | 281.416 | 7.1530 |
> | lg05 | 72.46 | 289.861 | 7.5423 |
> | rs07 | 72.46 | 289.887 | 7.5434 |
> | tkf91 | 73.87 | 295.506 | 7.7942 |
>
> **Q6:** TKF92 assumes the stationary distribution over sequence lengths is a zero-altered geometric distribution. If $\mu < \lambda$, then the length probability is negative (which is not possible). In other words, for a stationary distribution to exist, sequences cannot increase into infinite length.

---

> > ### Author Rebuttal · Reviewer_wWsj · 2026-04-03
> >
> > I thank the authors for their response and for answering my questions in depth. Most of my concerns have been addressed and I am therefore inclined to slightly raise my score.
> >
> > However, the rebuttal does not address W2 (incohesive narrative of the paper), which is arguably my biggest concern. My sense is that a paper can only be as strong as its story, and while there are several sound contributions here I'm not sure how well they tie in together - it feels to me more like several weaker papers merged together than one strong paper. For this reason I would not go higher than Weak Accept.
> >
> > Note that the author's response to Q1 mistakenly refers to W2, so I'd like to give them another chance to respond to this point, although there may not be much that can be done.

---

> > > ### Author Response · Authors · 2026-04-08
> > >
> > > We thank the reviewer for continued engagement with our rebuttal. Addressing W2 more directly:
> > >
> > > *"The TKF-based hierarchical mixtures are neat, but seem mostly independent from the comparison between Basic neural and Neural TKF."*
> > >
> > > These approaches to extending the TKF92 indel model (hierarchical mixtures, Basic neural, and the hybrid Neural TKF) are united by a common mathematical framework. Basic neural operates on a more concise alphabet of alignment columns compared to the other models, but it still satisfies the “alignment-Markovian” property. Furthermore, developing Basic neural and the hierarchical mixtures directly informed the subsequent hybrid approach, Neural TKF.
> > >
> > > To demonstrate the connection between Neural TKF and the hierarchical mixtures, we note that these all reduce to the reference TKF92+F81 implementation under appropriate parameterization.
> > >
> > > *"While there are several sound contributions here I'm not sure how well they tie in together."*
> > >
> > > Following this discussion and our discussion with Reviewer TzkA (particularly around their Q2, L2, W2, W3), we will update our camera-ready version to: 1.) make the connection between all models more explicit (especially the shared reduction of neural TKF and the hierarchical mixtures), and 2.) refine our conclusion to "celebrate the hybrid approach" (as suggested by Reviewer TzkA).

---

### Official Review · Reviewer_TLkk · 2026-03-11

**Soundness:** 3
**Presentation:** 3
**Significance:** 3
**Originality:** 3
**Overall Recommendation:** 4
**Confidence:** 3

**Summary:**

This work advances the field of phylogenetic indel modeling by developing a family of nested, analytically tractable models grounded in the TKF92 framework. By incorporating mixtures at three distinct biological levels—individual sites, contiguous fragments, and broader functional domains—the authors construct a richer probabilistic structure that is both alignment-Markovian and representable as a Pair-HMM with closed-form finite-time transition probabilities. Building on this foundation, the paper also introduces two neural approaches: a basic autoregressive model that sequentially predicts alignment columns conditioned on an ancestral sequence and divergence time, and a more sophisticated Neural TKF hybrid that combines learned sequence embeddings with the core TKF92+F81 mechanisms. Empirical evaluation on large, carefully curated pairwise alignments from Pfam reveals that a compact domain-mixture TKF92 model, with approximately 32,000 parameters, achieves test negative log-likelihood scores competitive with neural models containing 30 to 40 million parameters. The overall best performance, however, is attained by the Neural TKF model when powered by a six-block Transformer architecture.

**Compliance With Llm Reviewing Policy:**

Affirmed.

**Key Questions For Authors:**

1.  **Objective Alignment:** How exactly are training objectives aligned? Do HMMs maximize marginal likelihood (via Forward) while neural models optimize $P(A|X, t)$ on observed alignments? Have you tried training neural models against marginal likelihood surrogates?
2.  **Substitution Model Isolation:** Can you provide results isolating the effect of the substitution model (F81 vs. GTR-LG08) within each family to confirm robustness?
3.  **Indel Rate Context:** Are rates a function of observable sequence context (flanking residues) or solely mediated by latent fragment/domain classes?
4.  **Baselines Sensitivity:** How sensitive are the "Basic" models to architectural depth/width and time encoding? Could a tuned Basic model close the performance gap?
5.  **NLL vs. ECE Discrepancy:** Can you provide stratified analyses (by length, divergence, Pfam family) and calibration plots to explain the ranking differences?
6.  **Computational Demand:** How do mixture models compare to neural baselines regarding training time and GPU memory? Do they scale without the quadratic burden of attention?
7.  **Reproducibility:** Will you release code and data splits? If not, can you provide exhaustive hyperparameter details in an appendix?

**Limitations:**

yes

**Strengths And Weaknesses:**

## Strengths

1. The paper makes a notable theoretical contribution by extending the TKF92 framework to incorporate hierarchical nesting and mixtures at the site, fragment, and domain levels. The authors provide rigorous derivations for the resulting Pair-HMM transition probabilities, including a clever handling of non-emitting state summations via matrix inversion. They also introduce the concept of "alignment-Markovian" modeling, which offers a unifying lens to connect classical HMM-based CTMCs with modern neural autoregressive approaches. Among the neural extensions, the Neural TKF hybrid is particularly compelling—it embeds a theoretically grounded indel model into a neural representation learning pipeline, effectively combining inductive bias with data-driven flexibility.

2. The empirical evaluation is thorough and well-executed. The authors curated a large-scale dataset from Pfam with careful split strategies based on clans and families to minimize data leakage. They compare against a wide range of baselines—from classical models like TKF91 and TKF92 to more recent methods such as LG05 and RS07, as well as neural architectures including CNNs, LSTMs, and Transformers. By reporting both total negative log-likelihood and per-column Expected Calibration Error, the paper offers a nuanced view of model performance and highlights how metric choice can influence model ranking. The results convincingly show that compact, principled CTMC models can achieve log-likelihoods competitive with much larger neural networks while using orders of magnitude fewer parameters.

3. The exposition of the mathematical framework—particularly the step-by-step construction of mixture models and the derivation of transition and emission probabilities—is largely clear and accessible. More importantly, the paper delivers a strong message: theoretically grounded models with hierarchical structure can rival or outperform parameter-heavy neural alternatives in likelihood-based evaluation. This finding is significant for the field, as it challenges the assumption that larger neural models are inherently superior and encourages further exploration of hybrid or interpretable architectures.

## Weaknesses

1. One of the paper's central claims—that this is the "first HMM-based indel model to allow indel rates to depend on local sequence context"—is somewhat overstated. The proposed mechanism appears to achieve context dependence indirectly through latent mixture classes rather than through explicit functions of observable sequence features like flanking residues. Additionally, while the model conditions on the full ancestral sequence, its autoregressive formulation for the descendant sequence introduces an asymmetry; long-range dependencies and epistatic interactions are only implicitly captured, which may limit its biological realism.

2. The experimental design raises several concerns that complicate interpretation. The "Basic" autoregressive neural head may not have received the same level of hyperparameter tuning as the more complex Neural TKF model, potentially underrepresenting its true performance. Comparisons are further muddied by the use of inconsistent substitution models—GTR-LG08 in some experiments and F81 in others—making it difficult to assess NLL differences on equal footing. The evaluation would also benefit from additional analyses, such as empirical gap length distributions, robustness to branch-length misspecification, and alignment quality metrics. Moreover, the training objectives vary across model classes, with neural models optimizing conditional likelihood P(A|X, t) and HMMs optimizing joint likelihood P(A, X|t), which complicates direct comparison.

3. While the main text is generally clear, some derivations in the appendix—particularly those involving z_t and z_0—are overly terse and difficult to follow. A schematic diagram illustrating the "explode-and-sum" construction would greatly aid comprehension. The paper also misses opportunities to engage with recent relevant literature, such as work on epistasis in phylogenetic inference and theoretical advances in alignment under indel models. Incorporating these references would help better contextualize the contributions and limitations of the proposed approach.

---

> ### Author Rebuttal · Authors · 2026-03-31
>
> We thank the reviewer for the constructive feedback. We address these below, with W referring to weakness, Q referring to key questions.
>
> **W1, Q3:** *“The [claim] ‘first HMM-based indel model to allow indel rates to depend on local sequence context’ is somewhat overstated. The proposed mechanism achieves context dependence indirectly through latent mixture classes rather than through explicit functions of observable sequence features. [...] Long-range dependencies and epistatic interactions are only implicitly captured.”*
>
> It is possible to construct a model that directly incorporates long-range dependencies and epistatic interactions via context-dependent substitutions and indels. Such a model would be extremely difficult to solve. Existing phylogenetics approaches have limited success in modeling local substitution rate variation or incorporating indels (let alone rate-varying indels). Pragmatic approximations are required; introducing latent variables to decouple intractably-coupled dependencies is one such approach.
>
> Our approach is the first to yield a closed-form description of adjacency correlations, though we can see how our original claim may seem “overstated”. In the camera-ready version (CRV), we will clarify that our contribution represents a principled approximation to a hard problem rather than its exact solution.
>
> **W2, some Q1:**
>
> *“Hyperparameter tuning of the “Basic” neural models”*
>
> For all neural models, preliminary experiments showed that models were more sensitive to sequence embedder architecture than the architecture of the prediction heads. Thus, we formally perform hyperparameter sweeps with neural TKF, then transfer architecture designs to the Basic neural models. Basic neural models may benefit from their own hyperparameter tunings, but more intentional architecture designs (such as better representations of time) are likely to impact fit to data more.
>
> *“...empirical gap length distributions, robustness to branch-length misspecification, and alignment quality metrics”*
>
> We can provide empirical gap length distributions in the CRV.
>
> Regarding branch-length misspecification: In preliminary studies, we evaluated the likelihood $P(A \mid X)$ after marginalizing over a grid of times. While this extra summation makes our models robust to branch-length misspecification, doing so did not change rankings. Furthermore, $P(A \mid X)$ was close in value to likelihoods from conditioning on Pfam times, $P(A \mid  X, t)$. We concluded that marginalizing over a grid of times was not worth the extra computation. We can provide these results in the CRV.
>
> Could the reviewer clarify what alignment quality metrics they are referring to?
>
> *On aligning training objectives*
>
> Our hierarchical mixture models sum over all possible class labels at every column. This marginalization is easier if we also jointly infer the probability of the ancestor. Assuming our ancestral sequence also evolves as a similar birth-death process over fragments, we can 1. train models to maximize $P(A, X \mid t)$, 2. evaluate $P(X)$ with inferred parameters, and 3. divide to calculate $P(A \mid X, t)$. This allows direct comparison between models.
>
> **W3:** We will revise the appendix for clarity and include a detailed literature review in the CRV.
>
> **Rest of Q1:**  For details on training, see our response to Reviewer 4, wWsj (Q2 and Q3)
>
> **Q2:** For a comparison of basic indel models with the F81 substitution model, see response to Reviewer 4, wWsj (Q5)
>
> **Q4:** *“How sensitive are the ‘Basic’ models to architectural depth/width and time encoding?”*
>
> Width was moderately important, but depth provided the greatest boost to model fit to data. We have not evaluated different time encodings, but this would also likely affect model fit. A highly optimized ‘Basic’ model could close the performance gap with Neural TKF; this would be an interesting result!
>
> **Q5:**
> Stratifying results from Table 2 has yielded interesting insights; we thank the reviewer for this suggestion. Briefly, we find that the mixture of domain classes model exceeds on long sequences and long branch lengths. In fact, for the 5% most distant alignments in the test set (12,000 pairs), this model beats Neural TKF, as measured by exponential cross-entropy (ECE).
>
> | model | ECE |
> |---|---|
> | Mixture of domain classes, 10 comps. | 16.7652 |
> | Neural TKF, 6-block transformer | 17.4799 |
> | Neural TKF, LSTM | 17.5746 |
> | Neural TKF, CNN | 18.3745 |
> | Neural TKF, 1-block transformer | 18.3913 |
> | Basic neural, 6-block transformer | 18.8895 |
> | Basic neural, LSTM | 19.1817 |
> | Basic neural, CNN | 19.2894 |
> | ...fragment classes, 30 comps. | 19.5709 |
> | ...site classes, 30 comps. | 19.8936 |
> | Basic neural, 1-block transformer | 21.1555 |
>
> Could the reviewer clarify what they mean by calibration plots?
>
> **Q6:** For training times, see our discussion with Reviewer 4, wWsj (Q3).
>
> **Q7:** Yes, a public release of our code, training data, and model weights are forthcoming.

---

> > ### Author Rebuttal · Reviewer_TLkk · 2026-04-06
> >
> > I decide to keep my positive ratings.

---

### Official Review · Reviewer_cXgB · 2026-03-13

**Soundness:** 2
**Presentation:** 3
**Significance:** 2
**Originality:** 3
**Overall Recommendation:** 3
**Confidence:** 3

**Summary:**

the paper explores a fundamental challenge in biology: how to accurately model the way protein sequences change over millions of years. Traditional methods usually look at simple letter swaps in the DNA code but struggle with "indels," which are the messy insertions and deletions that change a sequence's length. The authors introduce a new mathematical framework that nests multiple "birth-death" processes inside one another. This allows the model to simulate not just single letter changes, but the evolution of entire groups of amino acids at once. By doing this, they create a bridge between classic statistical biology and modern machine learning, aiming to provide a model that is both biologically meaningful and mathematically powerful.

**Compliance With Llm Reviewing Policy:**

Affirmed.

**Key Questions For Authors:**

whether this nested approach can be expanded to account for "structural constraints," where the model would know that certain mutations are impossible because they would cause the protein to unfold.

**Limitations:**

the model currently focuses on "point-to-point" evolution rather than describing the entire population of a species, which might miss some subtle evolutionary pressures.

**Strengths And Weaknesses:**

strength: its ability to match the performance of massive, "black-box" neural networks using much simpler, biologically grounded equations. While many modern AI models for proteins require billions of parameters and are hard to interpret, this nested model achieves similar accuracy in predicting protein evolution while remaining transparent. A major highlight is its superior performance on "long-range" evolution tasks, where the model successfully predicts how proteins will look after vast stretches of time, outperforming many standard deep learning tools. Additionally, the paper is very rigorous in its testing, using a wide variety of datasets from different protein families to prove that the model isn't just a "one-hit wonder." Finally, the framework is designed to be flexible; it can be plugged into existing phylogenetic trees used by biologists, making it a highly practical tool for real-world evolutionary research.


weakness: First, while the model has fewer parameters than a neural network, the "nested" math is still quite heavy and might be difficult for average biologists to understand or use without specialized software. Second, the current version of the model assumes that different parts of a protein evolve somewhat independently, which ignores the complex "3D folding" interactions where a change in one corner of a protein might force a change in another distant corner. Third, the computational cost of calculating these nested probabilities can grow very quickly as protein sequences get longer, which might make the model too slow for analyzing massive "metagenomic" datasets that contain millions of long sequences. Fourth, the paper primarily compares its model against specific types of neural networks, and it remains to be seen if it can hold its own against the absolute newest "foundation models" for proteins that have been released in just the last few months.

---

> ### Author Rebuttal · Authors · 2026-03-31
>
> We thank the reviewer for constructive feedback and thoughtful ideas. We address concerns below, with W referring to weakness, Q referring to key questions, and L referring to limitations.
>
> **W1:** *“The ‘nested’ math [...] might be difficult for average biologists to understand or use without specialized software”*
> We recognize that descriptions of stochastic processes may not be for everybody. Although they are a necessary part of presenting this work, we have tried to make the descriptions accessible. We will continue to revise the camera-ready version for clarity. As for tool usage, we agree entirely; making this software user-friendly is a priority. An exciting future direction is to package these models into user-friendly tools that abstract away the underlying mathematical structures.
>
> **W2:** *“The current version of the model [...] ignores the complex ‘3D folding’ interactions where a change in one corner of a protein might force a change in another distant corner”*
> See our discussion with Reviewer 3, TLkk (W1).
>
> **W3:** *“The computational cost of calculating these nested probabilities [...] might make the model too slow for analyzing massive "metagenomic" datasets that contain millions of long sequences”*
> We agree with the reviewer: long sequences are harder to work with, which is part of why we have focused on proteins rather than genomes in our initial efforts. In principle, these methods could be extended to other sequences such as genomes. Developing the new components and methodologies that would be required to scale efficiently to such sequence lengths is an active field of research that parallels the ongoing efforts to extend context length in natural language models.
>
> For hierarchical mixture models with “nested probabilities”, where likelihood maximization requires marginalizing over class labels with the Forward algorithm, future implementations could replace the current sequential scan with a parallel prefix scan. This would make better use of GPU parallelism. Future neural models could leverage specialized architectures like Mamba (Gu and Dao 2024), which can tractably capture long-range dependencies.  Preparation of cleaned and curated training data would significantly increase the time required for such an effort.
>
> Overall, extending our methods to genomes would represent an important and exciting new line of model development!
>
> **W4:** *“It remains to be seen if [this model] can hold its own against the [...] newest foundation models for proteins that have been released in just the last few months.”*
>
> As a starting point for this comparison, we provide metrics from evaluating our test set with the left-to-right autoregressive likelihoods of Progen2 (Nijkamp et al 2023).
>
> | Metric | Value |
> |:---:|:---:|
> | Total negative log-likelihood (NLL, $\times 10^6$) | 74.44 |
> | Average NLL | 303.549 |
> | ECE | 8.9328 |
>
> Comparing these likelihoods, $P(Y)$, against ours, $P(A \mid X,t)$, isn’t exact, since we jointly model the likelihood of descendant $Y$ and alignment state path $\tau(A)$. Still, we note that even the simplest F81-TKF91 model achieves better fit to data than Progen2. This suggests that evolutionary context (i.e. conditioning on a known sequence and elapsed time) could be highly informative for sequence modeling. We are interested in exploring this further by comparing against the newest open-source, autoregressive PLMs.
>
> Additionally, we plan to enter our best models to the ProteinGym benchmark (Notin et al 2023). This would allow us to compare against the newest foundation models (as well as Potts models and classical phylogenetic methods) on the biologically relevant task of variant effect prediction.
>
> **Q:** *“whether this nested approach can be expanded to account for ‘structural constraints’, where the model would know that certain mutations are impossible”*
> For hierarchical mixture models (i.e. “nested approaches”), a natural extension to our methods would be to explicitly derive transition and emission matrices that account for co-evolution and epistasis, though this would require substantial derivations and analytical work. A quicker approach for incorporating structural constraints would be to augment sequence representations in the neural TKF models with the Foldseek 3di alphabet (van Kempen et al 2024), which encodes local structural context at each residue."
>
> **L:** *“the model currently focuses on "point-to-point" evolution rather than describing the entire population of a species”*
> This alludes to interesting future applications in population genetics. Specifically, mutation rates from these models could be used to build a more realistic simulation of evolution, which the field of population genetics depends on (Jerome Kelleher et al 2016). We would be open to collaborating with specialists in population genetics on more realistic simulations of evolution, particularly models of species-level evolution.

---

> > ### Author Rebuttal · Reviewer_cXgB · 2026-04-03
> >
> > I checked the rebuttal for this and other. Reviewers. While most points are appreciated I think 3 and 4 are still not directly resolved. I tend to keep my score

---

> > > ### Author Response · Authors · 2026-04-08
> > >
> > > We thank the reviewer for continued engagement with our rebuttal.
> > >
> > > *"W3 is not directly resolved"*
> > >
> > > We note that our proposed extensions for longer genomic sequences (parallel prefix scan for hierarchical mixture models, Mamba for neural models) are GPU-friendly and scale linearly with sequence length.
> > >
> > > *"W4 is not directly resolved"*
> > >
> > > We note that our neural models can be extended to incorporate foundation model embeddings, such as those from ESM (Hayes et al 2025).
> > >
> > > In the camera-ready version, a "future works" section (as suggested by Reviewer wWsj) will highlight these proposed extensions, among others suggested throughout this review process.

---

### Official Review · Reviewer_TzkA · 2026-03-13

**Soundness:** 3
**Presentation:** 4
**Significance:** 3
**Originality:** 4
**Overall Recommendation:** 4
**Confidence:** 4

**Summary:**

This paper seeks to study an important concept: how to accurately model protein sequence evolution over time, with a focus on insertions, deletions (indels), and structural heterogeneity. Overall, the main issue analyzed by the manuscript is whether continuous-time Markov chain (CTMC) models, specifically hierarchical extensions of the TKF92 model, can perform competitively against modern neural networks. The authors design several nested mixture models (e.g., DomMix) and a hybrid model called "Neural TKF". By training and testing on the Pfam dataset, they show that a pure statistical model with only 29K parameters achieves highly competitive negative log-likelihood (NLL) compared to basic neural networks with around 40M parameters. Moreover, their hybrid Neural TKF model achieves the best overall performance.

**Compliance With Llm Reviewing Policy:**

Affirmed.

**Final Justification:**

I recommend accepting this paper because the authors have addressed my concerns clearly in the rebuttal. They agreed to change the narrative to focus on the hybrid model, which clarifies their main contribution. The mathematical method is solid and the parameter efficiency is very impressive. With these revisions, I believe this work will be valuable for the field.

**Key Questions For Authors:**

1. **Regarding the paper's framing:** Why do you define a 43M parameter model as "parameter-heavy"? To make the paper more appealing to the ICML community, would you consider changing the narrative to focus on "the power of injecting evolutionary inductive biases into neural networks" (which your Neural TKF proves), rather than framing it as classical statistics versus neural networks?
2. **Regarding the conclusion:** Your Neural TKF model clearly wins in Table 2. Can you explain why the Discussion section does not highlight this as the primary technical contribution? I strongly suggest rewriting parts of the conclusion to celebrate this hybrid approach.
3. **Regarding the computational cost of latent variables:** In Section 5.1, you mention that unobserved fragment boundaries in TKF92 are a "minor technical inconvenience." Since they are latent variables that must be marginalized, does this marginalization algorithm become a computational bottleneck when you apply it to much longer sequences?
4. **Regarding text clarity:** Please check some phrasing for better readability. For example, in Section 5.3, you write that the 10-component mixture "ranks third overall". It would be clearer if you explicitly name the top two models in that sentence so readers do not have to guess from the table. Also, please review Section 3.1.2 for minor singular/plural grammar issues to ensure the mathematical descriptions are perfectly clear.

**Limitations:**

1. The neural network baselines are trained from scratch and do not utilize modern self-supervised pre-training, which limits the fairness of the comparison in real-world bioinformatics applications.
2. The pure hierarchical mixture models suffer from "component collapse" when scaling up, indicating a strict limit on their expressivity.
3. The evaluation depends entirely on curated Pfam pairwise alignments. It is not fully clear if the exact mathematical assumptions will remain stable on highly divergent or messy unaligned sequence data.

**Strengths And Weaknesses:**

**Strengths:**
1. **Impressive parameter efficiency:** The results of the 10-component domain mixture model are mathematically elegant. Achieving an NLL of 62.14 with only 29.23K parameters, compared to a 40M parameter neural network, strongly proves that using evolutionary first principles provides an excellent inductive bias.
2. **The hybrid approach is highly effective:** The proposed "Neural TKF" model is a very good idea. By using neural networks to predict site-specific parameters for the TKF92+F81 equations, the authors successfully combine the representation power of deep learning with strict physical interpretability.
3. **Solid mathematical foundation:** The exact derivations of the transition matrices for the nested CTMC mixtures (shown in the Appendix) are rigorous. The authors demonstrate strong skills in classical probabilistic modeling.

**Weaknesses:**
1. **The baseline comparison is not entirely fair for a modern ML venue:** The title and abstract criticize "parameter-heavy neural networks." However, a 43M-parameter Transformer is considered a very small model in 2026. The paper does not compare against large-scale pre-trained protein language models (PLMs). Therefore, the claim that these models can replace or beat modern deep learning is a bit overstated.
2. **The narrative contradicts the main experimental results:** According to Table 2, the Neural TKF model (with a 6-block Transformer) achieves the best Total NLL (61.72). This is a great success for hybrid architectures! However, the Discussion section seems to minimize this victory and spends too much text defending the 29K pure mixture model. This makes the logic of the paper confusing.
3. **Misinterpreting model capacity:** In Section 5.2, the authors state that the mixture models "never overfit" and that adding more components leads to "component collapse". From a machine learning perspective, this is a weakness, not a strength. It means the pure statistical model has a very low capacity ceiling and cannot scale to learn more complex data patterns.

---

> ### Author Rebuttal · Authors · 2026-03-31
>
> We thank the reviewer for the constructive feedback. We address concerns below, with W referring to weakness, Q referring to key questions, and L referring to limitations.
>
> **Q1, W1:** *“Why do you define a 43M parameter model as ‘parameter-heavy’?”*
>
> We agree that 43M is small in comparison to modern ML. Our choice of words was intended to emphasize that the classical models are tiny in comparison. We do see the value of refining the narrative in the camera-ready version, including in the title and abstract, to emphasize “the power of injecting evolutionary inductive biases into neural networks” (as suggested).
>
> **Q2, L2, W2, W3:** *“I strongly suggest rewriting parts of the conclusion to celebrate this hybrid approach. [...] The authors state that mixture models ‘never overfit’ and that adding more components leads to ‘component collapse’. From a machine learning perspective, this is a weakness, not a strength. It means the pure statistical model has a very low capacity ceiling and cannot scale to learn more complex data patterns.”*
>
> Related to refining our narrative, this particular combination of comments was especially helpful. As summarized and quoted here, the hierarchical mixture models are anticipated to hit a capacity ceiling. Additionally, neural TKF had the best fit to data in our model evaluation and has many promising avenues for further development (like using pre-trained ESM (Hayes et al 2025) embeddings). In the camera-ready version, we will update our discussion section to draw attention to the benefits of this hybrid approach, especially in light of the anticipated limitations of the hierarchical mixture models (as suggested).
>
> **Q4:** We will review Sections 5.3 and 3.1.2 as suggested. We also plan a careful close review of the camera-ready version for clarity. We greatly appreciate this detailed feedback!
>
> **Q3:** *“Since [unobserved fragment boundaries] are latent variables that must be marginalized, does this marginalization algorithm become a computational bottleneck?”*
>
> The original TKF92 HMM (Thorne et al 1992) and all our hierarchical mixture models have closed-form transition equations that fold in exact marginalization over latent fragment boundaries. For example, the TKF92 insert $\rightarrow$ insert transition probability is $r + (1−r) \beta$, where $r$ is the probability of extending the current inserted fragment and $(1−r) \beta$ is the probability of starting a new fragment that is also an insertion. All transition matrices include these sums and share this property. Thus, marginalizing over unknown fragment boundaries imposes no additional computational cost.
>
> **L1:** *“The neural network baselines are trained from scratch and do not utilize modern self-supervised pre-training.”*
>
> All neural networks are trained from scratch to enable a controlled comparison of model design choices, like sequence embedding architectures. It would be technically straightforward and likely advantageous to extend our framework to incorporate pretrained protein language model (PLM) embeddings, such as ESM. We agree with the reviewer; this would be an exciting next step in model development!
>
> **L3:** *“The evaluation depends entirely on curated Pfam pairwise alignments. It is not fully clear if the exact mathematical assumptions will remain stable on highly divergent or messy unaligned sequence data.”*
>
> Pfam pairwise alignments with short branch lengths (as opposed to highly divergent pairs) are advantageous for two reasons:
> 1. Fewer mutation events occur along short evolutionary trajectories, making them most informative for parameterizing instantaneous mutation rates. Longer branch lengths would add noise.
> 2. We do not realign pairs after extracting them from MSAs. For closely related sequences, the MSA-induced pairwise alignment is likely already optimal.
>
> All our models are alignment-Markovian, so alignments can be summed out using the Forward algorithm, or maximized with Viterbi; posterior counts can be obtained using Forward-Backward. For the HMM-based models, this is entirely standard. For the neural models, the Forward and Viterbi emissions and transitions are being calculated by neural networks, but the alignment-Markovian structure still allows summing over alignment paths. This is how similar neural transducers are trained (Graves 2012). The main complication is that doing so requires $O(L^2)$ time, which considerably slows training.
>
> Working with highly divergent, unaligned sequences would be a natural next step for evaluating our methods. Curating such a dataset (including a homology search pipeline to identify pairs) and training models on unaligned sequences would be exciting ways to extend our methodology. This would broaden applicability to the more challenging sequence pairs commonly encountered in bioinformatics (for example, proteins inferred from metagenomics datasets). We thank the reviewer for this suggestion.

---

> > ### Author Rebuttal · Reviewer_TzkA · 2026-04-01
> >
> > I have decided to keep my positive rating.

---

### Decision · Program_Chairs · 2026-04-30

**Decision:**

Accept (regular)

**Comment:**

This paper extends the TKF92 model of protein evolution by introducing nested hierarchical mixtures of continuous-time Markov chains to model insertions and deletions (indels). The authors demonstrate that these biologically grounded, parameter-efficient statistical models, alongside a novel neural-hybrid counterpart (Neural TKF), perform competitively with standard neural network architectures. Reviewers unanimously praised the mathematical rigor of the derivations, the elegant "alignment-Markovian" formulation, and the impressive parameter efficiency demonstrating the value of evolutionary inductive biases.

During the review process, reviewers raised concerns regarding the paper's narrative framing, noting that the phrasing unnecessarily pitted classical statistics against neural networks rather than celebrating the highly effective hybrid approach. Reviewers also requested clarifications on evaluation metrics, biological assumptions (such as time-reversibility), and the lack of comparisons against modern large-scale protein language models.
In rebuttal, the authors agreed to reframe the manuscript to center on the success of injecting evolutionary biases into neural networks. They provided new stratified analyses showing their pure mixture model excels on highly divergent alignments, justified their biological assumptions via Felsenstein’s pulley principle, and supplied a preliminary comparison demonstrating superior negative log-likelihood against Progen2. While one reviewer maintained a negative score due to concerns about the model's immediate scalability to massive metagenomic sequences and 3D folding constraints, the consensus recognized these as future application challenges rather than fatal flaws in the current methodology. Given the strong theoretical foundation, the rigorous empirical validation, and the highly promising hybrid architecture, the merits of the work confidently outweigh its limitations. We recommend acceptance.